# Causal Structure Learning in Hawkes Processes with Complex Latent Confounder Networks

**Songyao Jin**
University of California San Diego
soj007@ucsd.edu

**Biwei Huang**
University of California San Diego
bih007@ucsd.edu

## ABSTRACT

Multivariate Hawkes process provides a powerful framework for modeling temporal dependencies and event-driven interactions in complex systems. While existing methods primarily focus on uncovering causal structures among observed subprocesses, real-world systems are often only partially observed, with latent subprocesses posing significant challenges. In this paper, we show that continuous-time event sequences can be represented by a discrete-time causal model as the time interval shrinks, and we leverage this insight to establish necessary and sufficient conditions for identifying latent subprocesses and the causal influences. Accordingly, we propose a two-phase iterative algorithm that alternates between inferring causal relationships among discovered subprocesses and uncovering new latent subprocesses, guided by path-based conditions that guarantee identifiability. Experiments on both synthetic and real-world datasets show that our method effectively recovers causal structures despite the presence of latent subprocesses.

## 1 INTRODUCTION

Understanding causality in complex systems is essential across diverse scientific and practical domains, including social networks (Zhou et al., 2013), neuroscience (Bonnet et al., 2022), and finance (Hawkes, 2018). Multivariate Hawkes processes (Hawkes, 1971), with their ability to model temporal dependencies and event-driven interactions, have emerged as powerful tools for capturing these dynamics. A majority of existing approaches (Xu et al., 2016; Salehi et al., 2019; Idé et al., 2021) learns Hawkes processes by using Granger causality (Kim et al., 2011) and fitting continuous-time event sequences via maximum likelihood (Veen & Schoenberg, 2008). Another line of work reduces reliance on high-resolution event timestamps by performing likelihood-based estimation directly on pre-binned counts (Shlomovich et al., 2022; Cai et al., 2022; Qiao et al., 2023).

Almost all existing methods, including the two lines above, implicitly assume *causal sufficiency*: all task-relevant subprocesses (i.e., event sequences) are fully observed, and the goal is to uncover causal structure *only* among those observed subprocesses. In practice this assumption is often violated: many real-world systems are only partially observable, with some event sequences entirely unmeasured. For example, in neuroscience, limitations of neural recording leave many neurons unobserved even though they influence recorded neurons (Huang, 2015), obscuring the true causal structure. Identifying such latent subprocesses is crucial for reliable causal discovery, particularly when they act as confounders. Even though latent subprocesses cannot be directly intervened upon, failing to account for them can create spurious causal edges among observed subprocesses and lead to incorrect conclusions about the system's dynamics. Shelton et al. (2018) proposes a method to impute *missing* event times within an observed subprocess via posterior sampling, but this does not handle *entirely unobserved* subprocesses unless one specifies their existence and number in advance, an impractical requirement.

In this paper, we target the general scenario of *partial observability*: we seek to recover the causal structure among both observed and latent subprocesses *without prior knowledge* about whether latent subprocesses exist, how many there are, or where they connect. We first show that, as the time interval shrinks, the multivariate Hawkes process admits a discrete-time linear causal representa-

tion. Leveraging second-order (cross-covariance) statistics of this representation, we show that the causal graph is virtually identifiable, without prior knowledge of latent subprocesses, when each latent subprocess has suitable observed surrogates. Our main contributions are:

- To the best of our knowledge, we provide the first principled framework that identifies latent subprocesses and recovers causal structure in continuous-time event sequences *without* prior knowledge of the existence or number of latent subprocesses.
- By showing that multivariate Hawkes processes can be represented by a linear causal model over discretized variables, we derive necessary and sufficient conditions for identifying latent subprocesses and inferring causal influences.
- We develop a two-phase iterative algorithm that alternates between causal-structure recovery and latent-subprocess discovery, using rank tests on cross-covariance matrices of *observed* discretized variables; it requires no prior knowledge of latent components.

## 2    RELATED WORK

Related work relevant to this paper are two areas: Hawkes processes and causal discovery. We briefly review the most relevant works here; a more detailed discussion is deferred to Appendix A.

**Hawkes Processes.** Hawkes processes provide a flexible framework for modeling temporal dependencies among events (Hawkes, 1971; Laub et al., 2015). Existing methods for learning Hawkes structures from continuous-time data predominantly rely on likelihood-based estimation, using either parametric kernels (e.g., exponential, power-law) (Xu et al., 2016) or nonparametric procedures (Lewis & Mohler, 2011), often combined with sparsity regularization (Zhou et al., 2013; Idé et al., 2021). The NPHC method (Achab et al., 2018) estimates integrated kernels via moment-matching, providing a nonparametric alternative. Recent compression-based methods formulate causal discovery as selecting the Hawkes network that minimizes an minimum description length (MDL) or minimum message-length (MML) criterion (Jalaldoust et al., 2022; Hlaváčková-Schindler et al., 2024). These approaches are effective when all subprocesses are observed, but they do not account for latent components. When only binned event counts are available, another line of work fits Hawkes models by maximizing likelihood over bin counts (Shlomovich et al., 2022; Cai et al., 2022; Qiao et al., 2023), but such methods again presuppose full observability. In contrast, although our framework also operates on discretized data, it departs from likelihood-based fitting: by leveraging the autoregressive representation of Hawkes processes, we exploit low-rank patterns in cross-covariances, thus enabling the identification of *latent subprocesses*. Unlike bin-count likelihood methods such as Shlomovich et al. (2022), which reconstruct the likelihood of binned counts without modeling structural relations among bins, our approach establishes a linear causal representation of the discretized Hawkes process. This structural correspondence is fundamentally different and is elaborated in Appendix A.1.

**Causal Discovery.** Existing causal discovery methods are primarily designed for settings where variables follow deterministic relations, rather than stochastic event-driven dynamics such as Hawkes processes. Classical approaches for i.i.d. data include constraint-based (Spirtes et al., 2001), score-based (Chickering, 2002), and functional approaches (Shimizu et al., 2006). Extensions to handle latent variables include rank-based methods that identify hidden structures in linear models (Huang et al., 2022; Dong et al., 2023). While these allow latent variables, their guarantees typically hold only up to equivalence classes and rely on restrictive structural and cardinality assumptions that are incompatible with Hawkes dynamics. For time-series data, LPCMCI (Gerhardus & Runge, 2020) adapts conditional-independence testing to temporal domains. However, it assumes weak autocorrelation and exogenous latent variables, assumptions that are violated in Hawkes processes where dense cross-lag dependencies and endogenous latent subprocesses naturally arise. Although our method also leverages second-order (rank) statistics, it differs in two key respects. First, it targets *subprocesses* in multivariate Hawkes systems rather than static variables and accommodates both endogenous and exogenous latent subprocesses. Second, it establishes identifiability through time-aware rank constraints specifically tailored to Hawkes dynamics, avoiding the infeasible assumptions underlying existing rank- or independence-test based approaches.

## 3    PARTIALLY OBSERVED MULTIVARIATE HAWKES PROCESS-BASED CAUSAL MODEL

This section introduces the causal modeling framework for partially observed Hawkes processes and establishes the key definitions that support the subsequent results on structure discovery.

## 3.1 MULTIVARIATE HAWKES PROCESS

A multivariate Hawkes process is a self-exciting point process modeling temporal dependencies among events via a set of counting subprocesses $\mathcal{N}_\mathcal{G} = \{N_i\}_{i=1}^l$, where $N_i(t)$ records the number of type-$i$ events up to time $t$ (Hawkes, 1971; Laub et al., 2015). For each $i \in \{1, \ldots, l\}$, the intensity of subprocess $N_i$ that governs the event-triggering behavior is:

$$\lambda_i(t) = \mu_i + \sum_{j=1}^l \int_0^t \phi_{ij}(t-s) \, dN_j(s), \tag{1}$$

where $\mu_i$ is the background intensity, and $\phi_{ij}(s) \geq 0, \forall s \in (0, \infty)$ is excitation function, which measures the decaying influence of historical type-$j$ events on the subsequent type-$i$ events and is piecewise continuous. (Strictly) stationarity requires the spectral radius of the influence matrix $\Phi \in \mathbb{R}^{l \times l}$ with entries $\Phi_{ij} = \int_0^\infty \phi_{ij}(s) ds$ to be less than one (Bacry & Muzy, 2016). A detailed exposition of Hawkes process is deferred to Appendix B. See also Fig. 1a for illustration.

We are interested in identifying, for each subprocess $N_i$, the minimal set of subprocesses $\mathcal{P}_\mathcal{G} \subseteq \mathcal{N}_\mathcal{G}$ such that $\lambda_i(t)$ depends only on the historical events of the subprocesses in $\mathcal{P}_\mathcal{G}$ and not on others. Formally, this corresponds to $\int_0^t \phi_{ij}(t-s) \, dN_j(s) > 0$ for each $N_j \in \mathcal{P}_\mathcal{G}$, and zero otherwise. In this case, $N_i$ is said to be *locally independent* (Didelez, 2008) of $\mathcal{N}_\mathcal{G} \backslash \mathcal{P}_\mathcal{G}$ given $\mathcal{P}_\mathcal{G}$.

## 3.2 MODEL DEFINITION

To formalize our framework, we define a graphical causal model for multivariate Hawkes processes, where nodes represent subprocesses and directed edges correspond to nonzero excitation functions. The goal is to recover the causal structure among both observed and latent subprocesses.

**Definition 3.1** (Partially Observed Multivariate Hawkes Process-based Causal Model (PO-MHP)). Let $\mathcal{G} := (\mathcal{N}_\mathcal{G}, \mathcal{E}_\mathcal{G})$ be a directed graph, where each node $N_i \in \mathcal{N}_\mathcal{G}$ represents a subprocess in a multivariate Hawkes process. A directed edge $E_{ij} \in \mathcal{E}_\mathcal{G}$ exists iff $\int_0^t \phi_{ij}(t-s) \, dN_j(s) > 0$. The node set $\mathcal{N}_\mathcal{G}$ consists of $p$ observed nodes $\mathcal{O}_\mathcal{G} := \{O_i\}_{i=1}^p$ and $q$ latent nodes $\mathcal{L}_\mathcal{G} := \{L_i\}_{i=1}^q$, which correspond to the observed and latent subprocesses, respectively.

The PO-MHP model naturally allows cycles and self-loops that are typically challenging to analyze (Claassen & Mooij, 2023). Two subprocesses $N_i$ and $N_j$ form a directed cycle if there exist directed paths from $N_i$ to $N_j$ and from $N_j$ back to $N_i$. Any subprocess $N_i$ has self loop if $\int_0^t \phi_{ii}(t-s) \, dN_i(s) > 0$. Furthermore, it allows for the presence of latent subprocesses, and directed edges may exist between any pair of subprocesses, whether observed or latent; both $N_i$ and $N_j$ may be either observed or latent. To the best of our knowledge, it is the first principled work to investigate such a general structure in Hawkes processes.

**Definition 3.2** (Cause and Effect). For any $N_i, N_j \in \mathcal{N}_\mathcal{G}$, if there exists a directed path from $N_i$ to $N_j$, then $N_j$ is said to be a *effect* of $N_i$, and $N_i$ is said to be a *cause* of $N_j$.

**Definition 3.3** (Parent-Cause Set). For $N_i \in \mathcal{N}_\mathcal{G}$, the minimal set $\mathcal{P}_\mathcal{G} \subseteq \mathcal{N}_\mathcal{G} \setminus \{N_i\}$ is called its *parent-cause set* if every directed path from nodes in $\mathcal{N}_\mathcal{G} \backslash \{N_i\}$ to $N_i$ passes through some node in $\mathcal{P}_\mathcal{G}$. In the special case where $N_i$ has a self-loop, it is also included in $\mathcal{P}_\mathcal{G}$.

**Proposition 3.4** (Parent-Cause Set and Local Independence). *Subprocess $N_i$ is* locally independent *(defined in Section 3.1) of $\mathcal{N}_\mathcal{G} \backslash \mathcal{P}_\mathcal{G}$ given $\mathcal{P}_\mathcal{G}$ if and only if $\mathcal{P}_\mathcal{G}$ is the parent-cause set of $N_i$ in $\mathcal{N}_\mathcal{G}$.*

Proposition 3.4 is the PO-MHP analogue of the local Markov property: even with cycles and self-loops, conditioning on the parent-cause set $\mathcal{P}_\mathcal{G}$ screens off $N_i$ from all other subprocesses.

## 4 STRUCTURE IDENTIFICATION IN PARTIALLY OBSERVED HAWKES PROCESSES

In this section, we formalize partially observed Hawkes processes in a discrete-time framework and develop rank-based tests on observed counts that enable identification of the summary causal graph, including relationships mediated by latent subprocesses.

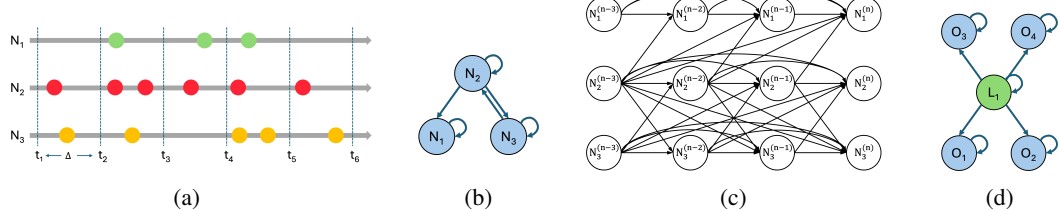

Figure 1: Figure 1: Illustration of multivariate Hawkes processes. (a) Point process representation with three subprocesses $N_1, N_2, N_3$, where the continuous timeline is partitioned into intervals of length $\Delta$. (b) The corresponding summary causal graph, the central object of this paper, with causal relations $N_1 \leftarrow N_2 \leftrightarrow N_3$ and self-loops on all nodes. (c) The window causal graph, showing the underlying time-lagged causal mechanism: each node denotes the count in one interval of length $\Delta$, modeled as a weighted sum of lagged parent nodes plus noise (Eq. 1). (d) A minimal example with a latent subprocess $L_1$ confounding $O_1$ and $O_2$, highlighting the primary focus of this paper.

### 4.1 FROM CONTINUOUS-TIME TO DISCRETE-TIME REPRESENTATION

Directly inferring the causal structure among subprocesses in a multivariate Hawkes process is challenging, particularly with latent subprocesses, as Eq. 1 defines a continuous-time stochastic process where $\lambda_i(t)$ is the expected history-dependent instantaneous rate. Instead of relying on conventional maximum-likelihood fitting, we present an explicit correspondence between Hawkes dynamics and a specified discrete-time linear autoregressive causal model, which enable the identification of latent subprocesses and the causal structure by applying statistical tests only on observed count data.

**Theorem 4.1** (Hawkes Process as a Linear Autoregressive Model). *Let $\mathcal{N}_{\mathcal{G}} := \{N_i\}_{i=1}^l$ be a stationary multivariate Hawkes process with background intensities $\{\mu_i\}_{i=1}^l$ and excitation functions $\{\phi_{ij}(s)\}_{i,j=1}^l$. Define the discretized event count in the $n$-th time window of size $\Delta \in (0, \delta)$ as*

$$N_i^{(n)} := N_i(n\Delta) - N_i((n-1)\Delta), \text{ with } N_i^{(0)} = 0,$$

*where $\delta > 0$ depends on the moment structure of the process. Then, as $\Delta \to 0$, the Hawkes process admits the linear autoregressive representation*

$$N_i^{(n)} = \sum_{j=1}^l \sum_{k=1}^n \theta_{ij}^{(k)} N_j^{(n-k)} + \varepsilon_i^{(n)} + \theta_i^{(0)}, \quad n \in \mathbb{Z}^+, \tag{2}$$

*where $\theta_i^{(0)} = \Delta \cdot \mu_i$ is the background parameter, $\theta_{ij}^{(k)} = \int_{(k-1)\Delta}^{k\Delta} \phi_{ij}(s) ds$ is the excitation coefficient, and $\varepsilon_i^{(n)}$ denotes the $n$-th realization of a serially uncorrelated white noise sequence.*

Theorem 4.1 shows that each current-bin count $N_i^{(n)}$ (referred to as *variable* hereafter) is a linear combination of lagged counts $\{N_j^{(n-k)}\}_{j \in \{1,\ldots,l\}}^{k \in \{1,\ldots,n\}}$ plus noise. The discretized variables therefore encode the causal structure of the underlying continuous-time subprocesses. As illustrated in Fig. 1, a directed edge $N_2 \to N_1$ in the summary graph corresponds to edges from all lagged variables of $N_2$ into $N_1^{(n)}$ in the window graph. This summary–window correspondence allows us to infer the causal structure among subprocesses by testing structural relations among the discretized variables. The proof of Theorem 4.1 appears in Appendix G.

In practice, it is unnecessary to include all lags $k = 1, \ldots, n$. Because the excitation function $\phi_{ij}(s)$, which serves as a decay kernel, typically has finite support, the coefficients $\theta_{ij}^{(k)} = \int_{(k-1)\Delta}^{k\Delta} \phi_{ij}(s) ds$ vanish for large $k$, distant lags $N_j^{(n-k)}$ therefore have little influence on $N_i^{(n)}$. We define the smallest cutoff $K$ at which this occurs as the number of *effective lags*. Accordingly, we truncate to at most $m$ lags with $m \geq K$, a standard finite-window practice in time-series estimation (Hyvärinen et al., 2010; Peters et al., 2013), though they assume acyclic summary graphs and understate latent components. A practical way to estimate $K$ is to retain lagged variables whose correlations with current variables remain statistically significant. Moreover, a bin width $\Delta$ small relative to the support of

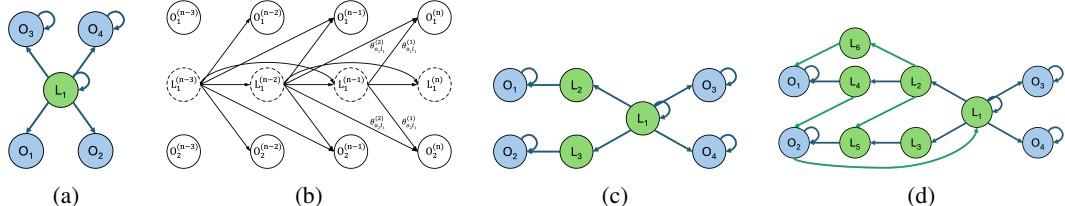

Figure 2: Examples of causal graphs with latent confounder subprocesses. (a) Summary graph where $O_1, O_2, O_3, O_4$ are observed and $L_1$ is latent. (Unlike Figure 1d, $O_1, O_2$ are shown without self-loops to simplify the derivation.) (b) Corresponding window causal graph among $O_1, O_2$, and $L_1$ with two effective lags. (c) More complex case where $L_1$ connects $O_1, O_2$ via intermediate latent subprocesses $L_2, L_3$. All subprocesses have self-loops except $L_2$ and $L_3$. (d) An even more intricate case, extending (c) with more complex intermediate latent subprocess paths and an additional edge $O_2 \to L_1$. All subprocesses except the intermediate latent ones have self-loops.

the excitation function is sufficient; see Appendix Q.3 for an empirical illustration. Data sparsity often offers intuitions: sparse event sequences correspond to excitation functions with larger effective support. In practice, we recommend selecting $\Delta$ via a simple grid search and choosing a value from the stable range, where the recovered structures remain robust to small perturbations in $\Delta$.

### 4.2 STRUCTURE DISCOVERY THROUGH RANK CONSTRAINTS

In this section, we link second-order statistics of Hawkes data to variables in the window causal graph, which in turn enables identification of the summary causal graph—even with latent subprocesses. Under the linear representation in Eq. 2 with white noise, the causal structure induces characteristic *low-rank* patterns in cross-covariance matrices of observed discretized variables.

#### 4.2.1 FROM OBSERVED PARENTS TO LATENT CONFOUNDER SUBPROCESSES

To illustrate, consider Fig. 1. Although the summary causal graph may contain directed cycles and self-loops, the associated window causal graph is a directed acyclic graph (DAG): by construction, future events cannot causally influence the past, reflecting the intrinsic temporal (Granger-causal) directionality (Shojaie & Fox, 2022). In this example, all three subprocesses are observed. In the summary graph, the parent-cause set of $N_1$ is $\mathcal{P}_\mathcal{G} \coloneqq \{N_i\}_{i \in \{1,2\}}$. In the window graph, consider the observed variable set $\mathbf{N}_v \coloneqq \{N_i^{(j)}\}_{i \in \{1,2,3\}}^{j \in \{n-m,\ldots,n\}}$, where $m$ is chosen to be at least the number of effective lags. Conditioning on the lagged variable set $\mathbf{P}_v \coloneqq \{N_i^{(j)}\}_{i \in \{1,2\}}^{j \in \{n-m,\ldots,n-1\}}$ renders $N_1^{(n)}$ d-separated from the remaining variables $\mathbf{R}_v \coloneqq \mathbf{N}_v \setminus (\mathbf{P}_v \cup N_1^{(n)})$. Consequently, the rank of the cross-covariance between $N_1^{(n)} \cup \mathbf{P}_v$ and $\mathbf{R}_v \cup \mathbf{P}_v$ equals $|\mathbf{P}_v|$. The theorems below formalize this connection between rank constraints and the window-graph structure, which we then leverage to identify the summary causal graph, including cases with latent subprocesses.

**Lemma 4.2** (D-separation and Rank Constraints in the Window Graph). *Consider the window causal graph of a PO-MHP. For any disjoint variable sets $\mathbf{A}_v$, $\mathbf{B}_v$ and $\mathbf{C}_v$, $\mathbf{C}_v$ d-separates $\mathbf{A}_v$ and $\mathbf{B}_v$, if and only if $\mathrm{rank}(\Sigma_{\mathbf{A}_v \cup \mathbf{C}_v, \, \mathbf{B}_v \cup \mathbf{C}_v}) = |\mathbf{C}_v|$, where $\Sigma_{\mathbf{A}_v \cup \mathbf{C}_v, \, \mathbf{B}_v \cup \mathbf{C}_v}$ denotes the **cross-covariance matrix** between $\mathbf{A}_v \cup \mathbf{C}_v$ and $\mathbf{B}_v \cup \mathbf{C}_v$, and $|\mathbf{C}_v|$ is the **cardinality** of $\mathbf{C}_v$.*

**Proposition 4.3** (Identifying Observed Parent-Cause Set). *Consider a PO-MHP with observed subprocesses $\mathcal{O}_\mathcal{G} \coloneqq \{O_i\}_{i=1}^p$. The followings are equivalent:*

- *In the summary graph, the set $\mathcal{P}_\mathcal{G} \subseteq \mathcal{O}_\mathcal{G}$ is the parent-cause set of the subprocess $O_1$.*
- *In the window graph, with the observed variable set $\mathbf{O}_v \coloneqq \{O_i^{(j)}\}_{i \in \{1,2,\ldots,p\}}^{j \in \{n-m,\ldots,n\}}$, $\mathcal{P}_\mathcal{G}$ is the minimal set such that lagged variable set $\mathbf{P}_v \coloneqq \{O_i^{(j)}\}_{O_i \in \mathcal{P}_\mathcal{G}}^{j \in \{n-m,\ldots,n-1\}}$ contains all parent variables of the current variable $O_1^{(n)}$.*
- *$\mathcal{P}_\mathcal{G}$ is the minimal set such that variable set $\mathbf{P}_v$ d-separates $O_1^{(n)}$ from the rest $\mathbf{O}_v \setminus \{\mathbf{P}_v \cup O_1^{(n)}\}$.*
- *$\mathcal{P}_\mathcal{G}$ is the minimal set such that $\mathrm{rank}(\Sigma_{O_1^{(n)} \cup \mathbf{P}_v, \, \mathbf{O}_v \setminus O_1^{(n)}}) = |\mathbf{P}_v|$.*

The criterion in Proposition 4.3 involves only observed variables. Whenever an observed subprocess $O_1$ satisfies this criterion, its parent-cause set $\mathcal{P}_{\mathcal{G}}$ is *uniquely determined* (by the minimality conditions in items 2–4), and $O_1$ is locally independent of all other observed subprocesses given $\mathcal{P}_{\mathcal{G}}$. As we show next, these four equivalent statements remain valid even with latent subprocesses.

One type of latent subprocess is the *intermediate latent subprocess*, which, if it exists, may lie on directed paths between an observed subprocess and each of its identified parent-cause set. In general, such intermediates are unidentifiable and typically omitted, since their effects can be absorbed by the identified parent-cause set. Nevertheless, owing to the specific structure of discretized Hawkes processes, once the observed parent causes of an observed subprocess have been identified via Proposition 4.3, it is possible to quantify the number of intermediate latent subprocesses along these paths. Detailed statements and proofs are deferred to Appendix C. Unless otherwise stated, for notational simplicity we do not consider intermediate latent subprocesses into parent-cause set.

Another type of latent subprocess is the *latent confounder subprocess*, a key focus of this paper. It is a latent subprocess that must be included in the parent-cause set to render its observed effect subprocess locally independent of others. For example, in Fig. 2a the structure $O_1 \leftarrow L_1 \rightarrow O_2$ makes $L_1$ a latent confounder of $O_1$ and $O_2$; in this case, $O_1$ and $O_2$ are locally independent only when conditioning on $L_1$. Proposition 4.3 will not identify the parent-cause set of either $O_1$ or $O_2$, since such confounders are unobserved. Furthermore, note that Eq. 1 does not specify the excitation function. We impose the following mild constraint on it to facilitate identification.

**Assumption 1** (Excitation Function). We consider that the excitation function takes the form $\phi_{ij}(s) = a_{ij}w(s)$, $\forall i, j \in \{1, \ldots, l\}$, where $a_{ij}$ is a constant capturing the peer influence between event types $i$ and $j$, and $w(s)$ is a common decay function depending only on the time lag $s$.

This assumption is quite general. For instance, the widely used exponential decay function $\alpha_{ij}e^{-\beta s}$ (Zhou et al., 2013) falls into this class. Other examples include normalized linear decay, normalized logistic decay, and related forms (Burt, 2000). Moreover, following standard practice in causal discovery (Spirtes, 2013; Huang et al., 2022), we also adopt the rank-faithfulness assumption for Hawkes processes. Intuitively, this assumption rules out pathological parameterizations where causal relationships cannot be identified. It holds generically with infinite data, since the degenerate cases where rank-faithfulness fails constitute a set of Lebesgue measure zero (Spirtes, 2013). Further details are deferred to Appendix D.

Intuitively, a latent confounder leaves a characteristic footprint in the second-order statistics of the observed subprocesses. When the same latent subprocess drives multiple observed subprocesses through the Hawkes excitation kernel, its contributions to their current values become aligned across time lags, so that the corresponding rows in suitable cross-covariance matrices lie in a low-dimensional subspace. This alignment manifests itself as a rank deficiency that cannot be explained by any purely observed parent-cause set. In the following, we make this intuition concrete by analyzing simple examples and then distilling the structural patterns into general rank-based conditions for identifying latent confounders.

### 4.2.2 CHARACTERIZING LATENT CONFOUNDERS VIA RANK CONSTRAINTS

Building on the above intuition, we now characterize when latent confounder subprocesses leave identifiable low-rank signatures in cross-covariance matrices, moving from illustrative examples to general conditions. Given the excitation function $\phi_{ij}(s) = a_{ij}w(s)$, the excitation coefficients in Eq. 2 are $\theta_{ij}^{(k)} = \int_{(k-1)\Delta}^{k\Delta} \phi_{ij}(s)\,ds = a_{ij}\int_{(k-1)\Delta}^{k\Delta} w(s)\,ds$, where the integral term $\int_{(k-1)\Delta}^{k\Delta} w(s)\,ds$ depends only on the time lag $k$. Consider the summary graph and its corresponding window graph with $m = 2$ lagged variables, as illustrated in Figs. 2a and 2b. According to the linear causal model in Eq. 2, the structural equations for the current variables $O_1^{(n)}$ and $O_2^{(n)}$ are

$$\begin{bmatrix} O_1^{(n)} \\ O_2^{(n)} \end{bmatrix} = \mathbf{E} \begin{bmatrix} L_1^{(n-1)} \\ L_1^{(n-2)} \end{bmatrix} + \begin{bmatrix} \epsilon_{o_1}^{(n)} + \theta_{o_1}^{(0)} \\ \epsilon_{o_2}^{(n)} + \theta_{o_2}^{(0)} \end{bmatrix}, \mathbf{E} = \begin{bmatrix} a_{o_1 l_1}\int_0^\Delta w(s)\,ds & a_{o_1 l_1}\int_\Delta^{2\Delta} w(s)\,ds \\ a_{o_2 l_1}\int_0^\Delta w(s)\,ds & a_{o_2 l_1}\int_\Delta^{2\Delta} w(s)\,ds \end{bmatrix}. \quad (3)$$

It is evident that the coefficient matrix $\mathbf{E}$ has rank 1. Consequently, the cross-covariance matrix satisfies $\text{rank}\left(\Sigma_{\{O_1^{(n)}, O_2^{(n)}\}, \{O_i^{(j)}\}_{i \in \{3,4\}}^{j \in \{n-m,\ldots,n\}}}\right) = 1$. For details, see Proposition 4.5 and its proof in Appendix K. This indicates a single latent confounder subprocess $L_1$ that serves as the parent cause of both $O_1$ and $O_2$, such that, conditional on $L_1$, $\{O_1, O_2\}$ are locally independent of $\{O_3, O_4\}$.

However, if $O_1$ and $O_2$ in Figure 2a also have self-loops (as illustrated in Figure 1d), the rank of the coefficient matrix is no longer 1. The self-loops generate additional indirect causal effects propagated from the lagged latent variables $\{L_1^{(j)}\}_{j \in \{n-m,\ldots,n-1\}}$ through the lagged observed variables $\{O_i^{(j)}\}_{i \in \{1,2\}}^{j \in \{n-m,\ldots,n-1\}}$ to the current variables $O_1^{(n)}$ and $O_2^{(n)}$. Since these lagged observed variables are available, we can include them in the cross-covariance matrix, which yields

$$\text{rank}\left(\Sigma_{\{O_i^{(j)}\}_{i \in \{1,2\}}^{j \in \{n-m,\ldots,n\}}, \{O_i^{(j)}\}_{i \in \{3,4\}}^{j \in \{n-m,\ldots,n\}} \cup \{O_i^{(j)}\}_{i \in \{1,2\}}^{j \in \{n-m,\ldots,n-1\}}}\right) = 2m+1,$$ where $2m$ corresponds

to the lagged variables of the two observed subprocesses $O_1$ and $O_2$, and 1 corresponds to the latent confounder subprocess. See Appendix E for further details.

Furthermore, in more complex scenarios, as illustrated in Figs. 2c and 2d, the causal pathways from the latent confounder $L_1$ to the observed subprocesses $O_1$ and $O_2$ become increasingly intricate. To address such cases, we formalize the *symmetric path situation*, which precisely characterizes those graphical configurations that induce rank deficiency in certain sub-covariance matrices of the observed subprocesses. It underpins the subsequent theorems by establishing a one-to-one correspondence between the underlying graph structure and its observable statistical properties.

**Definition 4.4** (Symmetric Acyclic Path Situation). Consider a latent confounder $L_1$ and an observed effect subprocess set $\mathcal{O}_{\mathcal{G}1}$. The following conditions define the *symmetric path situation*:

- There exist directed paths from $L_1$ to each subprocess in $\mathcal{O}_{\mathcal{G}1}$ such that each path consists exclusively of intermediate latent subprocesses (i.e., no observed subprocesses appear along these paths), and neither $L_1$ nor any subprocess in $\mathcal{O}_{\mathcal{G}1}$ appears as a non-end point along these paths.
- All such directed paths have the same length, meaning they contain the same number of intermediate latent subprocesses.
- All such directed paths are acyclic. Naturally, none of the intermediate latent subprocesses involved have self-loops.

The structure in Fig. 2c satisfies Definition 4.4, where the latent confounder $L_1$ connects $O_1$ and $O_2$ through the intermediate latent subprocesses $L_2$ and $L_3$, respectively, both without self-loops. If one intermediate subprocess is removed (e.g., $L_3$), the condition in Definition 4.4 is violated, since the path from $L_1$ to $O_1$ would then include one intermediate latent subprocess $L_2$, whereas the path from $L_1$ to $O_2$ would include none. Similarly, in the more complex structure shown in Fig. 2d, the core structure formed by the blue causal edges satisfies Definition 4.4, and the addition of the green edges still preserves this property. However, adding an extra edge, for instance, from $L_5$ to $L_3$, would break the condition, as it would introduce both asymmetry and cycles into the paths. The subsequent theorems will leverage this path condition to formally characterize graph structures involving the identification of latent confounder subprocesses.

**Proposition 4.5** (Identifying Latent Confounder from Observed Effects). *Consider a PO-MHP with excitation function $\phi_{ij}(s) = a_{ij}w(s)$ under rank faithfulness. The system consists of observed subprocesses $\mathcal{O}_{\mathcal{G}} := \{O_i\}_{i=1}^p$ and potentially latent subprocesses. Let $\mathbf{O}_v := \{O_i^{(j)}\}_{i \in \{1,\ldots,p\}}^{j \in \{n-m,\ldots,n\}}$ denote the set of corresponding observed variables. For any two observed subprocesses $\{O_1, O_2\}$,*

$$\text{rank}\left(\Sigma_{\{O_i^{(j)}\}_{i \in \{1,2\}}^{j \in \{n-m,\ldots,n\}}, \mathbf{O}_v \setminus \{O_1^{(n)}, O_2^{(n)}\}}\right) = 2m+1,$$ *if and only if there exists a latent confounder*

*subprocess $L_1$ in the parent-cause set of $\{O_1, O_2\}$ such that conditioning on $\mathcal{P}'_{\mathcal{G}} := L_1 \cup \{O_i\}_{i \in \{1,2\}}$ renders $\{O_1, O_2\}$ locally independent of nonempty set $\mathcal{O}_{\mathcal{G}} \setminus \mathcal{P}'_{\mathcal{G}}$, and $L_1$ with $\{O_1, O_2\}$ satisfy the Definition 4.4.*

Proposition 4.5 allows us to infer the existence of a latent confounder from its observed effects. This naturally raises the question: *How can we systematically infer the remaining causal relations involving these inferred latent subprocesses?* This challenge is illustrated in the four summary graphs of Fig. 3. In the following, we show how the observed effects can serve as surrogates for their latent confounders, thereby enabling recovery of the remaining causal structure.

**Definition 4.6** (Observed Effects as Surrogates). For each latent subprocess $L_1$ inferred from its observed effects $\{O_1, O_2\}$, we designate one of its observed effects, denoted $\mathcal{D}e(L_1) := O_1$, as an *observed surrogate* of $L_1$. The surrogate is chosen such that there exists a directed path from $L_1$ to $\mathcal{D}e(L_1)$ that does not traverse any other observed subprocess. We further define $\mathcal{S}ib(\mathcal{D}e(L_1))$ as the set of *observed siblings* of $\mathcal{D}e(L_1)$, consisting of all known other observed subprocesses influenced by $L_1$ through paths that likewise do not pass through any other observed subprocess.

For any observed subprocess $O_1$, we adopt the unified notation $\mathcal{D}e(O_1) = O_1$ and, correspondingly, $\mathcal{S}ib(\mathcal{D}e(O_1)) = \emptyset$. Moreover, $\mathcal{S}ib(\mathcal{D}e(L_1))$ captures the minimal set of observed subprocesses required to isolate the local influence of $L_1$ on the rest of the system, except through $\mathcal{D}e(L_1)$.

**Theorem 4.7** (Identifying Parent-Cause Set with Latent Confounder Involved). *Consider a PO-MHP with excitation function $\phi_{ij}(s) = a_{ij}w(s)$ under rank faithfulness. The system $\mathcal{N}_{\mathcal{G}} := \mathcal{O}_{\mathcal{G}} \cup \mathcal{L}_{\mathcal{G}}$ consists of observed subprocesses $\mathcal{O}_{\mathcal{G}} := \{O_i\}_{i=1}^p$, and inferred latent confounder processes $\mathcal{L}_{\mathcal{G}}$ whose parent-cause*

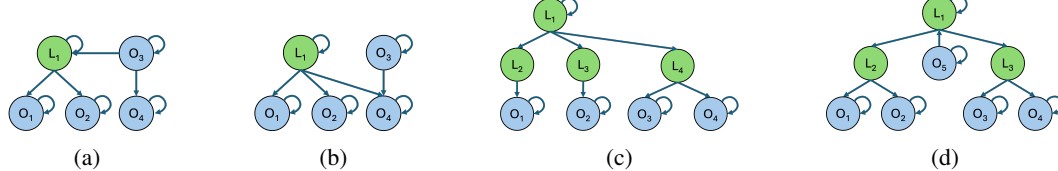

(a)        (b)        (c)        (d)

Figure 3: Illustrative examples of interactions among inferred latent confounder and the remaining observed subprocesses. In (a)–(c), assume $L_1$ has been inferred from its observed effects $\{O_1, O_2\}$. (a) $O_3$ causes $L_1$. (b) Both $L_1$ and $O_3$ cause $O_4$. (c) $L_1$ causes $L_4$, where $L_4$ can be inferred from $\{O_3, O_4\}$. (d) $L_1$ serves as the latent confounder of both inferred latent confounder $L_2$ and $L_3$.

sets remain to be identified. Let $\mathbf{O}_v \coloneqq \{O_i^{(j)}\}_{i \in \{1,\dots,p\}}^{j \in \{n-m,\dots,n\}}$ denote the corresponding observed variable set. For a subprocess $N_1 \in \mathcal{N}_{\mathcal{G}}$ and a candidate parent-cause set $\mathcal{P}'_{\mathcal{G}} \subseteq \mathcal{N}_{\mathcal{G}}$, when either $N_1$ is latent, or $\mathcal{P}'_{\mathcal{G}}$ contains latent subprocesses, or both, the following condition holds: $\mathcal{P}'_{\mathcal{G}}$ is the minimal set such that $\operatorname{rank}(\Sigma_{\mathbf{A}_v, \mathbf{B}_v}) = |\mathbf{A}_v| - 1$, where $\mathbf{A}_v \coloneqq \{\mathcal{D}e(N_1)^{(j)}, \mathcal{D}e(L_i)^{(j)}\}_{L_i \in \mathcal{P}'_{\mathcal{G}}}^{j \in \{n-m,\dots,n\}} \cup \{O_i^{(j)}\}_{O_i \in \mathcal{P}'_{\mathcal{G}}}^{j \in \{n-m,\dots,n-1\}} \cup \{O_i^{(j)}\}_{O_i \in \mathcal{S}ib(\mathcal{D}e(N_1)) \cup \{\mathcal{S}ib(\mathcal{D}e(L_i))\}_{L_i \in \mathcal{P}'_{\mathcal{G}}}}^{j \in \{n-m,\dots,n\}}$ and $\mathbf{B}_v \coloneqq \mathbf{O}_v \setminus \left(\mathcal{D}e(N_1)^{(n)} \cup \{\mathcal{D}e(L_i)^{(n)}\}_{L_i \in \mathcal{P}'_{\mathcal{G}}}\right)$, if and only if $\mathcal{P}'_{\mathcal{G}}$ is a subset of the true parent-cause set of $N_1$ such that: (i) conditioning on $\mathcal{S}_{\mathcal{G}} \coloneqq \mathcal{P}'_{\mathcal{G}} \cup \mathcal{D}e(N_1) \cup \{\mathcal{D}e(L_i)\}_{L_i \in \mathcal{P}'_{\mathcal{G}}} \cup \mathcal{S}ib(\mathcal{D}e(N_1)) \cup \{\mathcal{S}ib(\mathcal{D}e(L_i))\}_{L_i \in \mathcal{P}'_{\mathcal{G}}}$ renders $N_i$ locally independent of nonempty set $\mathcal{N}_{\mathcal{G}} \setminus \mathcal{S}_{\mathcal{G}}$; (ii) for each $L_i \in \mathcal{P}'_{\mathcal{G}}$, the latent confounder $L_i$ with observed effects $\{\mathcal{D}e(N_1), \mathcal{D}e(L_i)\}$ satisfies Definition 4.4; and (iii) all possible observed surrogates of $N_i$ in $\mathcal{O}_{\mathcal{G}}$ have been identified and added into the observed sibling set.

With Theorem 4.7 (and Proposition 4.3), we can identify arbitrary causal relations among both observed and inferred latent subprocesses. This naturally raises a final question: *How can we further discover new latent subprocesses that are causally related to inferred latent subprocesses, as in Fig. 3d?* As shown below, the observed surrogate can still be leveraged for this purpose.

**Theorem 4.8** (Identifying Latent Confounder from Latent Confounder ). *Consider a PO-MHP with excitation function $\phi_{ij}(s) = a_{ij}w(s)$ under rank faithfulness. The system $\mathcal{N}_{\mathcal{G}} \coloneqq \mathcal{O}_{\mathcal{G}} \cup \mathcal{L}_{\mathcal{G}}$ consists of observed subprocesses $\mathcal{O}_{\mathcal{G}} \coloneqq \{O_i\}_{i=1}^{p}$, and inferred latent confounder subprocesses $\mathcal{L}_{\mathcal{G}}$ whose parent-cause sets remain unidentified by Theorem 4.7. Let $\mathbf{O}_v \coloneqq \{O_i^{(j)}\}_{i \in \{1,\dots,p\}}^{j \in \{n-m,\dots,n\}}$ denote the corresponding observed variable set. For any two subprocesses $N_1, N_2 \subseteq \mathcal{N}_{\mathcal{G}}$ (either observed or latent), $\operatorname{rank}(\Sigma_{\mathbf{A}_v, \mathbf{B}_v}) = |\mathbf{A}_v| - 1$, where $\mathbf{A}_v \coloneqq \{\mathcal{D}e(N_i)^{(j)}\}_{i \in \{1,2\}}^{j \in \{n-m,\dots,n\}} \cup \{O_i^{(j)}\}_{O_i \in \mathcal{S}ib(\mathcal{D}e(N_1)) \cup \mathcal{S}ib(\mathcal{D}e(N_2))}^{j \in \{n-m,\dots,n\}}$, and $\mathbf{B}_v \coloneqq \mathbf{O}_v \setminus \{\mathcal{D}e(N_1)^{(n)}, \mathcal{D}e(N_2)^{(n)}\}$, if and only if there exists a latent confounder subprocess $L_1$ in the parent-cause set of $\{N_1, N_2\}$ such that: (i) conditioning on $\mathcal{P}'_{\mathcal{G}} \coloneqq L_1 \cup \{N_i\}_{i \in \{1,2\}} \cup \{\mathcal{S}ib(\mathcal{D}e(N_i))\}_{i \in \{1,2\}}$ renders $\{N_1, N_2\}$ locally independent of nonempty set $\mathcal{N}_{\mathcal{G}} \setminus \mathcal{P}'_{\mathcal{G}}$; (ii) $L_1$ with $\{\mathcal{D}e(N_1), \mathcal{D}e(N_2)\}$ satisfies Definition 4.4; and (iii) all possible observed surrogates of $\{N_1, N_2\}$ in $\mathcal{O}_{\mathcal{G}}$ have been identified and added into the observed sibling set.*

Theorem 4.7 and Theorem 4.8 are extensions of Proposition 4.3 and Proposition 4.5, respectively. These extend the framework by replacing latent subprocesses with their observed surrogates when evaluating the rank of the relevant cross-covariance matrices. Equipped with these four key theorems, we are now ready to present the discovery algorithm in the next section.

## 5   RANK-BASED DISCOVERY ALGORITHM

In this section, we present a two-phase iterative algorithm that leverages the identification theorems to progressively recover causal relationships among discovered subprocesses and to uncover new latent subprocesses. Let $\mathcal{A}_{\mathcal{G}}$ denote the *active process set*, consisting of subprocesses whose parent causes remain unidentified. Initially, $\mathcal{A}_{\mathcal{G}}$ is set to the observed subprocess set $\mathcal{O}_{\mathcal{G}}$ and is updated throughout the procedure. Moreover, due to cycles in the summary causal graph, observed subprocesses previously identified as effects may still act as causes for other subprocesses in $\mathcal{A}_{\mathcal{G}}$, and thus remain under investigation. The overall procedure is in Algorithm 1.

**Phase I: Identifying Causal Relations**   Each iteration begins with Phase I, which identifies the causal structure of under-investigated subprocesses (both observed and latent) in $\mathcal{A}_{\mathcal{G}}$. In this phase, we systematically iterate over each subprocess in $\mathcal{A}_{\mathcal{G}}$ and test its parent causes using the current set $\mathcal{A}_{\mathcal{G}} \cup \mathcal{O}_{\mathcal{G}}$. If a subprocess's parent-cause set is fully contained by this set, it can be identified using Proposition 4.3 and Theorem 4.7. Once identified, the subprocess is removed from $\mathcal{A}_{\mathcal{G}}$. The phase continues until no further updates occur. Detailed steps are given in Algorithm 2 in Appendix O.1.

---

**Algorithm 1** Two-Phase Iterative Discovery Algorithm

---

**Input:** Observed subprocess set $\mathcal{O}_\mathcal{G}$
**Output:** Causal graph $\mathcal{G}$
1: Initialize partial causal graph $\mathcal{G} \coloneqq \emptyset$ and active process set $\mathcal{A}_\mathcal{G} \coloneqq \mathcal{O}_\mathcal{G}$.
2: **repeat**
3:     $(\mathcal{G}, \mathcal{A}_\mathcal{G}) \leftarrow$ Identifying Causal Relations $(\mathcal{G}, \mathcal{A}_\mathcal{G}, \mathcal{O}_\mathcal{G})$.                // Phase I
4:     $(\mathcal{G}, \mathcal{A}_\mathcal{G}) \leftarrow$ Discovering New Latent Subprocesses $(\mathcal{G}, \mathcal{A}_\mathcal{G}, \mathcal{O}_\mathcal{G})$.     // Phase II
5: **until** $\mathcal{A}_\mathcal{G}$ is empty or no updates occur.
6: **return** $\mathcal{G}$

---

**Phase II: Discovering New Latent Subprocesses**   When no additional subprocesses in $\mathcal{A}_\mathcal{G}$ can be resolved in Phase I, the algorithm proceeds to Phase II. Here, we search for new latent confounder subprocesses by exhaustively checking all pairs in $\mathcal{A}_\mathcal{G}$ using Proposition 4.5 and Theorem 4.8. Identified latent confounders are merged if pairs overlap in subprocesses, implying a shared latent parent cause. $\mathcal{A}_\mathcal{G}$ is then updated by adding the new latent subprocesses and removing their effects. After that, the algorithm returns to Phase I. This procedure continues until $\mathcal{A}_\mathcal{G}$ is empty or unchanged. Details are provided in Algorithm 3 in Appendix O.2.

**Theorem 5.1** (Identifiability of the Causal Graph). *Consider a PO-MHP with excitation function $\phi_{ij}(s) = a_{ij}w(s)$ under rank faithfulness. If every latent confounder subprocess, along with all its observed surrogates ($\geq 2$), satisfies Definition 4.4, then the causal graph over observed and latent confounder subprocesses can be identified. When no latent subprocesses exist, the causal graph is fully identifiable using only Phase I.*

Moreover, the computational complexity depends on the number of subprocesses (including latent confounders) and the density of the underlying causal graph, which together determine the number of iterations required for complete graph discovery. A detailed complexity analysis is in Appendix P.

## 6 EXPERIMENTS

**Synthetic Data**   We compare our method against six strong baselines. SHP (Qiao et al., 2023) and THP (Cai et al., 2022) are likelihood-based approaches for discretized (binned) Hawkes data, while NPHC (Achab et al., 2018) is a cumulant-based method for original Hawkes data. Since existing Hawkes-based methods cannot identify latent subprocesses without prior knowledge, we additionally include two rank-based approaches developed for i.i.d. linear latent models, Hier. Rank (Huang et al., 2022) and RLCD (Dong et al., 2023), as well as LPCMCI (Gerhardus & Runge, 2020), a time-series baseline that accounts for exogenous latent confounder variables, though all three rely on strong assumptions not satisfied in our setting (See Appendix A). For these methods, we apply them to discretized Hawkes data. For our method, we evaluate both event sequences generated by the Hawkes process in Eq. (1) and data generated directly from the discrete-time model in Eq. (2). We test on six synthetic graph families: the fully observed graph in Fig. 1b and five structures with latent subprocesses in Figs. 2a and 3a–3d. Results are reported as average F1-scores over ten runs on a personal CPU machine. Additional experimental details and extended results (larger graphs, sensitivity to $\Delta$, and robustness to rank-faithfulness violations) are provided in Appendix Q. As shown in Fig. 4, our method consistently outperforms all baselines. Notably, latent cases typically require larger sample sizes: because the spectral radius of a stationary Hawkes process is $< 1$, causal influences attenuate along latent paths, making reliable detection more data-demanding.

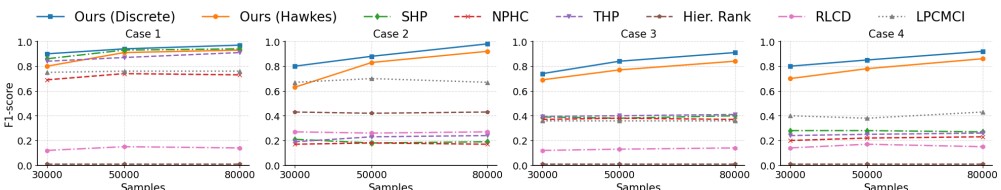

Figure 4: F1-score comparisons for first four synthetic causal graphs (Cases 1–4), corresponding to the structures in Figs. 1b, 2a, 3a and 3b. See Appendix Q.3 for additional cases.

**Real-world Data**   We evaluate our method on a public cellular network dataset (Qiao et al., 2023) with expert-validated ground truth. The original dataset contains 18 alarm types from 55 devices ($\approx$ 35k events over eight months), though not every device exhibits all alarms. We focus on `device_id=8`, which includes the alarms relevant to the subgraph of interest. For evaluation, we consider a five-alarm subgraph (`Alarm_ids=0-3` and 7) and treat `Alarm_id=7` as latent by manual exclusion. Notably, `Alarm_id=1` and

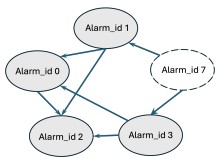

Figure 5: Inferred causal subgraph from the cellular network dataset, where `Alarm_id=7` is successfully identified as a latent subprocess.

`Alarm_id=3` are observed effects of this latent subprocess, enabling its recovery from observed data. Our inferred graph (Fig. 5) successfully recovers the latent subprocess and its primary influences. Furthermore, on this sub-dataset our method quantitatively outperforms representative baselines; see Appendix Q.4 for details.

## 7    CONCLUSION AND FUTURE WORK

We presented a principled framework for structure learning in partially observed multivariate Hawkes processes (PO-MHP), capable of uncovering both causal relationships among observed subprocesses and latent confounder subprocesses influencing them. By leveraging sub-covariance rank conditions and a tailored path constraint, we derived necessary and sufficient conditions for identifiability and designed a two-phase iterative algorithm that reconstructs the full causal graph with guarantees. One future research direction is to relax the excitation function, for example, by introducing node-specific decay rates, to broaden its applicability. Another direction is to develop discovery algorithms with lower complexity. Furthermore, more diverse real-world datasets can be applied to gain deeper domain insights.

## ACKNOWLEDGMENTS

The authors extend their sincere gratitude to the anonymous reviewers for their insightful feedback and constructive suggestions. Biwei Huang would like to acknowledge the support of the National Science Foundation (Grant No. DMS-2428058).

## ETHICS STATEMENT

This work makes a primarily methodological contribution, proposing a framework for structure learning in partially observed multivariate Hawkes processes. The study does not involve human subjects, personal or sensitive data, or applications with direct societal or ethical risks. Our experiments are conducted on synthetic data and a publicly available dataset with expert-validated ground truth. We therefore believe that the research does not raise ethical concerns beyond standard scientific integrity, and is fully consistent with the ICLR Code of Ethics.

## REPRODUCIBILITY STATEMENT

We have made significant efforts to ensure the reproducibility of our results. All theoretical results are stated with explicit assumptions, and full proofs are provided in the appendix. Experimental details, including data preprocessing steps, and additional results (e.g., sensitivity to $\Delta$ and robustness analyses), are included in the appendix and supplementary materials. Synthetic data generation procedures are fully specified, and the real-world dataset we use is publicly available.

## LLM USAGE STATEMENT

Large Language Models (LLMs) were used only as general-purpose tools to aid in polishing the writing and improving clarity of presentation. They were not involved in research ideation, methodological design, experiments, and analysis. No LLM qualifies for authorship.

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

# Appendix

## A  DETAILED RELATED WORK

We review related work in three areas: point processes, Hawkes processes, and causal discovery.

**Point Processes**   A large body of research studies temporal dependencies in general point processes. Meek (2014) introduced a graphical framework based on $\delta^*$-separation and process independence to connect graphical representations with statistical properties. Gunawardana et al. (2011) proposed a one-dimensional point process model with piecewise-constant conditional intensity, enabling closed-form Bayesian inference of temporal dependencies. Chwialkowski & Gretton (2014) developed a kernel-based independence test for general random processes, offering a nonparametric perspective.

Other works focus on specific structural settings. Basu et al. (2015) studied Granger causality for discrete transition processes with grouping structures. Daneshmand et al. (2014) proposed a parametric cascade generative process for continuous-time diffusion networks. In the setting of marked point processes, Didelez (2008) introduced graphical models capable of capturing local independence across marks, generalizing dependency analysis in complex systems.

**Hawkes Processes**   Hawkes processes (Hawkes, 1971; Laub et al., 2015) form a prominent class of self-exciting point processes that model how past events influence future occurrences. A major research line estimates Hawkes structures from continuous-time event data, typically via likelihood-based methods. These approaches assume parametric excitation kernels such as exponential or power-law (Zhou et al., 2013; Zhao et al., 2015), or adopt nonparametric procedures (Lewis & Mohler, 2011; Luo et al., 2015), often combined with sparsity or low-rank regularization (Xu et al., 2016; Idé et al., 2021). The NPHC method (Achab et al., 2018) provides a nonparametric alternative, estimating integrated kernels through moment-matching. While effective when all subprocesses are observed, these approaches do not address the presence of latent components.

When only discretized or binned counts are available, another line of work fits Hawkes models directly from bin data. Shlomovich et al. (2022) proposed an EM algorithm with importance sampling to recover parameters from binned observations without precise timestamps. Qiao et al. (2023) introduced SHP, which learns causal structure from binned event sequences using sparsity-regularized likelihood. Cai et al. (2022) developed THPs, which impose topological constraints to recover causal influences from discrete sequences. These binned-likelihood approaches, however, assume full observability and do not infer the existence or number of latent subprocesses.

**Causal Discovery**   Causal discovery aims to infer causal relations from data, traditionally under i.i.d. assumptions with DAG structures (Pearl, 2009). Classical approaches include constraint-based methods (e.g., PC Spirtes et al. (2001)), score-based methods (e.g., GES Chickering (2002)), and functional approaches (e.g., LiNGAM Shimizu et al. (2006)). Latent variables pose major challenges, leading to extensions such as FCI and its variants (Spirtes et al., 1995; Colombo et al., 2012; Claassen et al., 2013), which use conditional independence to infer partial structures under exogenous latent confounder variables.

More recent advances explore settings with causally related latent confounder variables. Huang et al. (2022) and Dong et al. (2023) identify equivalence classes in linear models using second-order (rank) statistics. However, these methods typically rely on restrictive assumptions—such as hierarchical latent structures or cardinality constraints—that are incompatible with Hawkes dynamics. In Hawkes-induced time series, autoregressive representations are dense across lags, observed surrogates are often fewer than effective latent "parents," and endogenous latent confounder subprocesses arise naturally. Other works, such as Xie et al. (2020; 2022) and Jin et al. (2023), leverage higher-order statistics to improve identifiability, but they still assume i.i.d. data, which may introduce spurious dependencies once temporal dynamics are ignored.

Causal discovery has also been extended to time-series domains. Approaches such as SVAR-based LiNGAM (Hyvärinen et al., 2010) and PC-style temporal methods like PCMCI (Runge, 2020) and LPCMCI (Gerhardus & Runge, 2020) rely on conditional independence tests over lagged variables. Yet these methods presuppose weak autocorrelation and exogenous latent variables, assumptions that are violated in Hawkes processes, which exhibit dense cross-lag interactions and endogenous latent subprocesses.

## A.1   DETAILED RELATION TO BINNED HAWKES ESTIMATION

Shlomovich et al. (2022) address parameter estimation for binned Hawkes processes via a modified EM algorithm when only bin counts $N_t = N((t+1)\Delta) - N(t\Delta)$ are observed and exact event times are unavailable. The bin counts are treated as observed data and the unobserved event times $\mathcal{T}$ as latent variables (their Eq. 6). Because direct Monte Carlo sampling of $\mathcal{T}$ is intractable in Hawkes models, they employ importance sampling to simulate within-bin timestamps that match the observed counts, thereby maximizing the (binned) likelihood (see their Sec. 2).

Our goal and methodology differ. Leveraging the link between INAR and linear autoregressive models, Theorem 4.1 establishes an explicit linear structural representation for discretized multivariate Hawkes processes. This connection enables causal discovery directly over binned variables—including the identification of latent confounder subprocesses—with identifiability guarantees (Propositions 4.3 and 4.5; Theorems 4.7 and 4.8). In contrast to likelihood maximization based on simulated event times, our framework uses time-aware rank

constraints on cross-covariances to recover causal structure. To the best of our knowledge, prior work has not provided a direct, theoretically grounded reduction from Hawkes processes to linear structural models for the purpose of causal discovery.

## A.2 DETAILED RELATION TO RANK-BASED LATENT DISCOVERY

Huang et al. (2022) (and related works by Xie et al. (2022) and Dong et al. (2023)) study latent structure discovery under i.i.d. assumptions and continuous variables. Our problem differs substantively: we aim to recover causal structure among *observed and latent subprocesses* in multivariate Hawkes processes, where each subprocess is a point process and inference is performed on discretized representations.

**Different Data Domain and Causal Assumptions**  Huang et al. (2022) (and Xie et al. (2022)) assume a latent hierarchical structure, specifically: (i) there are no direct causal links among observed variables, and all dependencies among observed variables arise exclusively from their latent confounder variables; and (ii) observed variables cannot cause latent variables, i.e., endogenous latent confounders are ruled out (see Eq. 1 and Definition 1 in Huang et al. (2022), and Eq. 1, 2 and Definition 1 in Xie et al. (2022)). Neither assumption is needed in our framework. We allow both direct observed-to-observed edges (see Proposition Proposition 4.3 in our paper) and the existence of endogenous latent confounder subprocesses that can be caused by observed subprocesses (see Theorem 4.7 in our paper).

**Cardinality Requirements vs. Hawkes Density**  Huang et al. (2022), Xie et al. (2022), and Dong et al. (2023) rely on a cardinality condition of the form |children| > |parents| for certain latent sets (cf. Definition 4 in Huang et al. (2022), Condition 1 in Xie et al. (2022), Definition 5 in Dong et al. (2023)). This is generally incompatible with discretized Hawkes processes, whose autoregressive representation is inherently dense (Eq. 2 in our paper): if a latent $L_1$ causes $O_2$, then each discretized variable $O_2^{(n)}$ is influenced by *many* lags of $L_1$ (potentially hundreds or thousands in practice), making the required |children| > |parents| condition fail systematically. Our method avoids such cardinality assumptions: leveraging the separable excitation (Assumption 1), we place lagged observed variables on both sides of carefully chosen cross-covariance blocks so that rank deficiency reliably signals latent confounders (lines 199–216; Proposition 4.5; Theorem 4.8).

**Time-Aware vs. i.i.d. Causal Discovery**  The above i.i.d. methods do not exploit temporal order and, in principle, can test variables at time $n$ as putative parents of variables at time $n - 1$. Our procedure is explicitly time-aware: candidate parents for $t = n$ are restricted to appropriate lags (Propositions 4.3 and 4.5; Theorems 4.7 and 4.8), aligning identification with Hawkes dynamics. This distinction mirrors PC (Spirtes et al., 2001) (i.i.d.) vs. PCMCI (Runge, 2020) (time series).

## B  MULTIVARIATE HAWKES PROCESS DETAILS

Before introducing multivariate Hawkes process, we first describe the temporal point process and counting process briefly. A temporal point process is a random process whose realization consists of a list of discrete events in time $\{T_1, T_2, \dots\}$ taking values in $[0, \infty)$. Another equivalent representation is the counting process, $N_1 = \{N_1(t) | t \in [0, \infty)\}$, where $N_1(t)$ records the number of events *before* time $t$ and $N_1(0) = 0$. A multivariate point process with $l$ types of events is represented by $l$ counting processes $\{N_i\}_{i=1}^l$ on a probability space $(\Omega, \mathcal{F}, \mathbb{P})$. $N_i = \{N_i(t) | t \in [0, \infty)\}$, where $N_i(t)$ is the number of type-$i$ events occurring *before* time $t$ and $N_i(0) = 0$. $\mathbf{U} = \{1, \dots, l\}$ (sometimes abbreviated as $[l]$) represents the set of event types. $\Omega = [0, \infty) \times \mathbf{U}$ is the sample space. $\mathcal{F} = \mathcal{F}(t)$ is a filtration, that is, a non-decreasing family of $\sigma$-algebras which for each time point $t \in \mathbb{R}$, represent the set of event sequences the processes can realize *before* time $t$. $\mathbb{P}$ is the probability measure. Point processes can be characterized by the conditional intensity function, which models patterns of interest, such as self-triggering or self-correcting behaviors (Xu et al., 2015). The conditional intensity function is defined as the expected instantaneous rate of type-$i$ events occurring at time $t$, given the event history:

$$\lambda_i(t) = \lim_{h \to 0} \frac{\mathbb{E}[N_i(t + h) - N_i(t) | \mathcal{H}(t)]}{h}, \tag{4}$$

where $\mathcal{H}(t) = \{(t_k, i) | t_k < t, i \in \mathbf{U}\}$ collects historical events of all types *before* time t. The multivariate Hawkes process is a class of multivariate point processes characterized by a self-triggering pattern as defined in Eq. 1.

## C  IDENTIFYING INTERMEDIATE LATENT SUBPROCESSES

As shown in the summary causal graph in Fig. 6a, $L_1$ is an intermediate latent subprocess on the directed path from the observed subprocess $O_2$ to $O_3$. According to Proposition 4.3, $L_1$ is not identifiable and its effect is

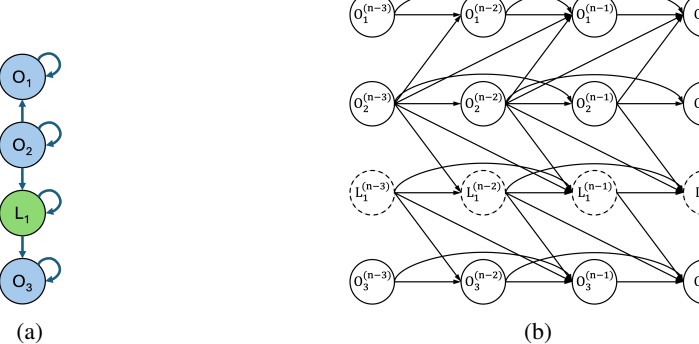

Figure 6: Example of an intermediate latent subprocess on the directed path from $O_2$ to $O_1$. (a) The summary causal graph, where $L_1$ is the intermediate latent subprocess. (b) The corresponding window causal graph with two effective lags.

attributed to $O_2$, leading to the inference that $O_2$ is the parent cause of $O_3$. This is because the influence of $L_1$ is indistinguishable from that of $O_2$ and can be effectively merged into $O_2$.

Consider now the corresponding window causal graph shown in Fig. 6b. The observed variable set is given by $\mathbf{O}_v \coloneqq \{O_i^{(j)}\}_{i\in\{1,2,3\}}^{j\in\{n-m,\ldots,n\}}$, where $m = 3$ exceeds the number of effective lags (which is 2 in this example). Instead of conditioning on all three lagged variables $\{O_2^{(n-1)}, O_2^{(n-2)}, O_2^{(n-3)}\}$ of $O_2$, we exclude $O_2^{(n-1)}$ and condition only on $\{O_2^{(n-2)}, O_2^{(n-3)}\}$. In this case, $O_3^{(n)}$ becomes d-separated from the remaining variables in $\mathbf{O}_v$. This property arises because, due to the presence of the intermediate latent subprocess $L_1$, $O_2^{(n-1)}$ no longer has a direct influence on $O_3^{(n)}$. The following corollary formalizes a general method for identifying the number of intermediate latent subprocesses that may exist between an observed subprocess and each of its inferred observed parent causes.

**Corollary C.1** (Identifying Intermediate Latent Subprocesses). *Let $\mathcal{O}_{\mathcal{G}} \coloneqq \{O_i\}_{i=1}^p$ denote the observed sub-processes, with the corresponding observed variable set $\mathbf{O}_v \coloneqq \{O_i^{(j)}\}_{i\in\{1,2,\ldots,p\}}^{j\in\{n-m,\ldots,n\}}$. Consider an observed subprocess $O_1$ and its inferred observed parent-cause set $\mathcal{P}_{\mathcal{G}} \subseteq \mathcal{O}_{\mathcal{G}}$. For any $O_2 \in \mathcal{P}_{\mathcal{G}}$, let $h$ be the largest value such that the lagged variable set $\mathbf{P}_v \coloneqq \{O_i^{(j)}\}_{O_i\in\mathcal{P}_{\mathcal{G}}}^{j\in\{n-m,\ldots,n-1\}} \setminus \{O_2^{(j)}\}_{j\in\{n-h,\ldots,n-1\}}$ d-separates $O_1^{(n)}$ from the remaining variables $\mathbf{O}_v \setminus \{\mathbf{P}_v \cup O_1^{(n)}\}$. Equivalently, $h$ is the largest value such that:*

$$\text{rank}\left(\Sigma_{\{O_1^{(n)}\}\cup\mathbf{P}_v,\, \mathbf{O}_v\setminus\{O_1^{(n)}\}}\right) = |\mathbf{P}_v|.$$

*This is equivalent to stating that the shortest directed path from $O_2$ to $O_1$ that does not pass through any other observed subprocess consists of $h$ latent subprocesses.*

*Remark* C.2. In Corollary C.1, $O_1$ and $O_2$ may refer to the same subprocess in cases where Proposition 4.3 infers that $O_1$ has a self-loop. In such cases, Corollary C.1 can be used to determine whether this self-loop represents a direct self-excitation or is mediated through intermediate latent subprocesses.

*Proof.* Let $\mathcal{O}_{\mathcal{G}} \coloneqq \{O_i\}_{i=1}^p$ and $\mathbf{O}_v \coloneqq \{O_i^{(j)}\}_{i\in\{1,2,\ldots,p\}}^{j\in\{n-m,\ldots,n\}}$. Consider an observed subprocess $O_1$ and its inferred parent-cause set $\mathcal{P}_{\mathcal{G}}$. For any $O_2 \in \mathcal{P}_{\mathcal{G}}$, assume the shortest directed path from $O_2$ to $O_1$ consists of $h$ latent subprocesses. This implies that the lagged variables $\{O_2^{(j)}\}_{j\in\{n-h,\ldots,n-1\}}$ do not influence $O_1^{(n)}$, while the variables $\{O_2^{(j)}\}_{j\in\{n-m,\ldots,n-h\}}$ do.

Thus, the variable set $\mathbf{P}_v = \{O_i^{(j)}\}_{O_i\in\mathcal{P}_{\mathcal{G}}}^{j\in\{n-m,\ldots,n-1\}} \setminus \{O_2^{(j)}\}_{j\in\{n-h,\ldots,n-1\}}$ is the minimal set that d-separates $O_1^{(n)}$ from the remaining variables. By Lemma 4.2, this implies:

$$\text{rank}\left(\Sigma_{\{O_1^{(n)}\}\cup\mathbf{P}_v,\, \mathbf{O}_v\setminus\{O_1^{(n)}\}}\right) = |\mathbf{P}_v|.$$

This completes the proof. □

## D  RANK FAITHFULNESS FOR THE HAWKES PROCESS

**Assumption 2** (Rank Faithfulness for the Hawkes Process). A probability distribution $p$ is rank faithful to the graph $\mathcal{G}$ if every rank constraint on any sub cross-covariance matrix that holds in $p$ is entailed by every linear structural model (as defined in Eq. 2) with respect to $\mathcal{G}$ and the excitation function $\phi_{ij}(s) = a_{ij}w(t)$, $\forall i, j \in \{1, \ldots, l\}$.

The rank faithfulness assumption is widely adopted in the causal discovery literature for i.i.d. data (Spirtes, 2013; Huang et al., 2022). In our setting, it concerns only the excitation function coefficients $a_{ij}$, and prior studies have shown that violations of this assumption occur only in degenerate cases of Lebesgue measure zero. Specifically, it fails only in rare pathological scenarios, such as when multiple $a_{ij}$ coefficients involving those of latent subprocesses are exactly equal across different subprocesses in a manner that induces rank deficiency—situations that are highly unlikely to arise in practical applications.

To empirically assess the robustness of our method to potential violations of rank faithfulness, we conduct a sensitivity analysis where, for each synthetic graph, we choose the exponential excitation function $\phi_{ij}(s) = \alpha_{ij}e^{-\beta s}$ and deliberately assign identical $a_{ij}$ values to two randomly selected edges, thereby artificially increasing the risk of the violation of rank faithfulness. The results, reported in Table 3 in Appendix Q.3, demonstrate that our method remains robust even under such perturbations.

## E  ACCOUNTING FOR SELF-LOOPED OBSERVED SUBPROCESSES UNDER LATENT CONFOUNDER INFLUENCE

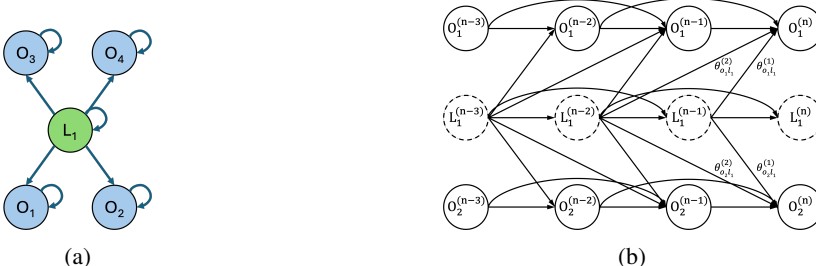

(a)             (b)

Figure 7: Illustration of self-Looped observed subprocesses under latent confounder influence. (a) Summary causal graph where $O_1, O_2, O_3$, and $O_4$ are observed subprocesses, and $L_1$ is a latent confounder subprocess. All subprocesses have self-loops. (b) Corresponding window causal graph for (a), illustrating the discretized causal mechanisms among $O_1, O_2$, and $L_1$, with two effective lags.

Consider Fig. 7, where $O_1$ and $O_2$ also have self-loops. As shown in Fig. 7b, these self-loops introduce additional indirect effects, where the lagged latent variables $\{L_1^{(j)}\}_{j \in \{n-m, \ldots, n-1\}}$ propagate their influence to the current variables $O_1^{(n)}$ and $O_2^{(n)}$ through the observed lagged variables $\{O_i^{(j)}\}_{i \in \{1,2\}}^{j \in \{n-m, \ldots, n-1\}}$.

Fortunately, since these lagged variables are observed, they can be explicitly incorporated into the structural equations and, correspondingly, into the covariance matrix. Considering the window graph in Fig. 7b with $m$ effective lagged variables, the structural equations for the observed variables $\{O_i^{(j)}\}_{i \in \{1,2\}}^{j \in \{n-m, \ldots, n\}}$ can be written as:

$$
\begin{bmatrix} O_1^{(n)} \\ O_1^{(n-1)} \\ \cdots \\ O_1^{(n-m)} \\ O_2^{(n)} \\ O_2^{(n-1)} \\ \cdots \\ O_2^{(n-m)} \end{bmatrix} = \mathbf{E} \begin{bmatrix} L_1^{(n-1)} \\ \cdots \\ L_1^{(n-m)} \\ O_1^{(n-1)} \\ \cdots \\ O_1^{(n-m)} \\ O_2^{(n-1)} \\ \cdots \\ O_2^{(n-m)} \end{bmatrix} + \begin{bmatrix} \epsilon_{o_1}^{(n)} + \theta_{o_1}^{(0)} \\ \epsilon_{o_1}^{(n-1)} + \theta_{o_1}^{(0)} \\ \cdots \\ \epsilon_{o_1}^{(n-m)} + \theta_{o_1}^{(0)} \\ \epsilon_{o_2}^{(n)} + \theta_{o_1}^{(0)} \\ \epsilon_{o_2}^{(n-1)} + \theta_{o_1}^{(0)} \\ \cdots \\ \epsilon_{o_2}^{(n-m)} + \theta_{o_1}^{(0)} \end{bmatrix},
\tag{5}
$$

$$
\mathbf{E} =
\begin{bmatrix}
a_{o_1 l_1} \int_0^\Delta w(s)ds & \cdots & a_{o_1 l_1} \int_{(m-1)\Delta}^{m\Delta} w(s)ds & 1 & \overset{1}{\cdots} & 1 & 0 & \overset{0}{\cdots} & 0 \\
0 & \overset{0}{\cdots} & 0 & 1 & \overset{0}{\cdots} & 0 & 0 & \overset{0}{\cdots} & 0 \\
\cdots & \cdots & \cdots & \cdots & \cdots & \cdots & \cdots & \cdots & \cdots \\
0 & \overset{0}{\cdots} & 0 & 0 & \overset{0}{\cdots} & 1 & 0 & \overset{0}{\cdots} & 0 \\
a_{o_2 l_1} \int_0^\Delta w(s)ds & \cdots & a_{o_2 l_1} \int_{(m-1)\Delta}^{m\Delta} w(s)ds & 0 & \overset{0}{\cdots} & 0 & 1 & \overset{1}{\cdots} & 1 \\
0 & \overset{0}{\cdots} & 0 & 0 & \overset{0}{\cdots} & 0 & 1 & \overset{0}{\cdots} & 0 \\
\cdots & \cdots & \cdots & \cdots & \cdots & \cdots & \cdots & \cdots & \cdots \\
0 & \overset{0}{\cdots} & 0 & 0 & \overset{0}{\cdots} & 0 & 0 & \overset{0}{\cdots} & 1
\end{bmatrix}
\left.\begin{matrix}\\\\\\\\\end{matrix}\right\}m \quad . \quad \left.\begin{matrix}\\\\\\\\\end{matrix}\right\}m
\tag{6}
$$

It is straightforward to see that the rank of the coefficient matrix $\mathbf{E}$ is $2m + 1$. Accordingly, by including these observed lagged variables in the cross-covariance matrix, we obtain:

$$
\mathrm{rank}\left(\Sigma_{\{O_i^{(j)}\}_{i \in \{1,2\}}^{j \in \{n-m,\dots,n\}}, \{O_i^{(j)}\}_{i \in \{3,4\}}^{j \in \{n-m,\dots,n\}} \cup \{O_i^{(j)}\}_{i \in \{1,2\}}^{j \in \{n-m,\dots,n-1\}}}\right) = 2m + 1,
$$

where $2m$ corresponds to the observed lagged variables of $O_1$ and $O_2$, and 1 corresponds to the latent confounder subprocess $L_1$. For a formal proof, see Proposition 4.5 and Appendix K. This result implies the presence of a latent confounder subprocess $L_1$, such that the set $\{L_1, O_1, O_2\}$ forms the parent-cause set of $\{O_1, O_2\}$. Conditioning on this set renders $\{O_1, O_2\}$ locally independent of $O_3$ and $O_4$.

## F   PROOF OF PROPOSITION 3.4

*Proof.* According to Definition 3.3, a set $\mathcal{P}_{\mathcal{G}1}$ is the parent-cause set for $N_i \in \mathcal{N}_\mathcal{G}$ if and only if every directed path from any node in $\mathcal{N}_\mathcal{G} \backslash \{N_i\}$ to $N_i$ passes through at least one node in $\mathcal{P}_{\mathcal{G}1}$. In the special case where $N_i$ has a self-loop, $N_i$ itself is also included as a parent cause. This implies that each subprocess $N_j \in \mathcal{P}_{\mathcal{G}1}$ must have a directed influence on $N_i$, formally expressed as $\int_0^t \phi_{ij}(t-s) \, dN_j(s) > 0$, and zero otherwise.

On the other hand, as discussed in the last paragraph of Section 3.1 and in (Didelez, 2008), $N_i$ is said to be *locally independent* of the rest of the system given a minimal set $\mathcal{P}_{\mathcal{G}2} \subseteq \mathcal{N}_\mathcal{G}$ if and only if $\int_0^t \phi_{ij}(t-s) \, dN_j(s) > 0$ for each $N_j \in \mathcal{P}_{\mathcal{G}2}$, and zero otherwise.

Therefore, we can see $\mathcal{P}_{\mathcal{G}1} = \mathcal{P}_{\mathcal{G}2}$. $N_i$ is *locally independent* of $\mathcal{N}_\mathcal{G} \backslash \mathcal{P}_\mathcal{G}$ given $\mathcal{P}_\mathcal{G}$ if and only if $\mathcal{P}_\mathcal{G}$ is the parent-cause set for $N_i$ in $\mathcal{N}_\mathcal{G}$. This completes the proof of Proposition 3.4. $\qquad\square$

## G   PROOF OF THEOREM 4.1

*Proof.* To prove Theorem 4.1, we proceed in three steps. First, we define the multivariate INAR sequence (Definition G.1) and show that it admits a linear autoregressive model representation (Proposition G.3). Then, in Theorem G.5, we establish that this multivariate INAR counting process converges weakly to a multivariate Hawkes process as the bin size $\Delta \to 0$, with the correspondence between the parameters of both models made explicit. The details are as follows:

**Step 1: Definition of the Multivariate INAR model.**   We begin by introducing the multivariate INAR model, adapted from Definition 20 in the paper B. Hawkes forests in (Kirchner, 2017).

**Definition G.1** (Multivariate integer-valued autoregressive model (Kirchner, 2017)). An multivariate integer-valued autoregressive time series(multivariate INAR) is a sequence of $\mathbb{N}_0$-valued random variables $\mathbf{X}_v = \{X_1^{(n)}, X_2^{(n)}, \dots, X_l^{(n)}\}_{n \in \mathbb{Z}^+}$ with $X_i^{(0)} = 0$, defined as:

$$
X_i^{(n)} = \sum_{j=1}^l \sum_{k=1}^n \sum_{h=1}^{X_j^{(n-k)}} \xi_h^{(\theta_{ij}^{(k)})} + \epsilon_i^{(n)}, \quad i \in \{1, \dots, l\}, n \in \mathbb{Z}^+,
\tag{7}
$$

where the reproduction coefficients $\theta_{ij}^{(k)} \geq 0$ with the subcritical matrix $[\sum_{k=1}^n \theta_{ij}^{(k)}]_{(i,j) \in \{1,\dots,l\}}$, and the immigration coefficients $\theta_i^{(0)} \geq 0$. $\epsilon_i^{(n)} \overset{iid}{\sim} \mathrm{Pois}(\theta_i^{(0)})$ and $\xi_h^{(\theta_{ij}^{(k)})} \overset{iid}{\sim} \mathrm{Pois}(\theta_{ij}^{(k)})$ are mutually independent and also independent of $\epsilon_i^{(n)}$.

*Remark* G.2. Definition G.1 follows Definition 20 in paper B. Hawkes Forests in (Kirchner, 2017), but with adapted notation to match Theorem 4.1. Key correspondences include: $d \to l$, $i \to j$, $j \to i$, $l \to h$,

$(\mathbf{X}_n)_{n\in\mathbb{Z}} \to \mathbf{X}_v$, $X_n^{(j)} \to X_i^{(n)}$, $\xi_{n,l}^{(i,j,k)} \to \xi_h^{(\theta_{ij}^{(k)})}$, $\epsilon_n^{(j)} \to \epsilon_i^{(n)}$, $\alpha_{i,j,k} \to \theta_{ij}^{(k)}$, $\alpha_{0,j} \to \theta_i^{(0)}$. We also restrict indices to $n \in \mathbb{Z}^+$ to match our Hawkes process formulation (Eq. 1); this is purely notational and does not affect the model semantics, as the indices are used to describe relative positions within the time series.

**Step 2: Linear autoregressive representation of the INAR model.** The multivariate INAR sequence admits an equivalent linear autoregressive representation, as shown in Proposition G.3, corresponding to Proposition 3.1 in (Kirchner, 2016a). The current variable $X_i^{(n)}$ is expressed as a weighted sum of all lagged variables $X_j^{n-k}$, plus a constant term $\theta_i^{(0)}$ and a stationary white-noise term $\varepsilon_i^{(n)}$.

**Proposition G.3.** *Let $\mathbf{X}_v$ be a l-dimensional INAR sequence as in Definition G.1 with immigration coefficients $\theta_i^{(0)} \geq 0$, reproduction coefficients $\theta_{ij}^{(k)} \geq 0$, and $X_i^{(0)} = 0$. Then*

$$\varepsilon_i^{(n)} := X_i^{(n)} - \theta_i^{(0)} - \sum_{j=1}^{l}\sum_{k=1}^{n} \theta_{i,j}^k X_j^{(n-k)}, \quad n \in \mathbb{Z}^+, \tag{8}$$

*defines a white-noise sequence, i.e., $(\varepsilon_i^{(n)})$ is stationary, $\mathbb{E}[\varepsilon_i^{(n)}] = 0$, $i \in \{1,\ldots,l\}$, $n \in \mathbb{Z}^+$. Moreover, let the $l \times l$ noise matrices $\mathbf{u}_n\mathbf{u}_{n'}^\top := [\varepsilon_i^{(n)}\varepsilon_j^{(n)}]_{(i,j)\in\{1,\ldots,l\}}$ and reproduction-coefficient matrices $A_k := [\theta_{ij}^{(k)}]_{(i,j)\in\{1,\ldots,l\}}$, we have:*

$$\mathbb{E}[\mathbf{u}_n\mathbf{u}_{n'}^\top] = \begin{cases} \mathrm{diag}\left(\left(I_{l\times l} - \sum_{k=1}^{n} A_k\right)^{-1}\right), & n = n', \\ 0_{l\times l}, & n \neq n'. \end{cases} \tag{9}$$

*Remark* G.4. Proposition G.3 is adapted from Proposition 3.1 of (Kirchner, 2016a), which also appears as Proposition 6 of the same paper in the author's doctoral thesis (Kirchner, 2017). The original formulation uses full vector and matrix notation; here, we present each dimension separately for consistency with our notation. Moreover, we adapted notations as in Remark G.2.

**Step 3: Convergence of the INAR to a Hawkes process.** Finally, we show that the multivariate INAR process converges to a multivariate Hawkes process as $\Delta \to 0$. The corresponding parameters of the INAR and the Hawkes process are also stated in the below theorem.

**Theorem G.5** (Multivariate INAR converging to multivariate Hawkes process (Kirchner, 2017)). *Let $\mathcal{N}_{\mathcal{G}1} = \{N_i\}_{i=1}^l$ be a stationary multivariate Hawkes process with background intensities $\{\mu_i\}_{i=1}^l$, and piecewise-continuous excitation functions $\{\phi_{ij}(s) \geq 0, \forall s \in (0,\infty)\}_{i=1}^l$. For bin width $\Delta \in (0,\delta)$, let $\mathbf{X}_v = \{X_1^{(n)}, X_2^{(n)}, \ldots, X_l^{(n)}\}_{n\in\mathbb{Z}^+}$ be an multivariate INAR sequence with:*

$$\theta_i^{(0)} = \Delta\mu_i, \quad \theta_{ij}^{(k)} = \int_{(k-1)\Delta}^{k\Delta} \phi_{ij}(s)ds,$$

*and $X_i^{(0)} = 0$. From the sequences $\mathbf{X}_v$, we define a family of point processes $\mathcal{N}_{\mathcal{G}2} = \{N_i^\Delta\}_{i=1}^l$, where for each $N_i^\Delta$,*

$$N_i^\Delta(t) := \sum_{n:n\Delta\leq t} X_i^{(n)}, \quad t \in [0,\infty). \tag{10}$$

*Then, $\mathcal{N}_{\mathcal{G}2}$ converges weakly to $\mathcal{N}_{\mathcal{G}1}$ in distribution, as $\Delta \to 0$.*

*Remark* G.6. Theorem G.5 is a simplified version of Theorem 25 in (Kirchner, 2017). The original proof proceeds via convergence of Hawkes forests (constructed via branching random walks), showing that the Hawkes process is a limit of INAR-based approximating forests. The convergence of Hawkes process and INAR comes from the convergence of Hawkes forest and the approximating forest with corresponding parameters. We adapt it here with a direct correspondence between Hawkes and INAR parameters, and restrict domains to $t \in [0,\infty)$ and $n \in \mathbb{Z}^+$ for consistency and clarification. Typically, Hawkes process results hold for both domains (Laub et al., 2015, Remark 2), since variable $t$ and $n$ is used only to calibrate relative positions. Moreover, besides the notation changes in Remark G.2, we adopt: $\mathbf{N_F} \to \mathcal{N}_{\mathcal{G}1}$, $\mathbf{N_{F(\Delta)}} \to \mathcal{N}_{\mathcal{G}2}$, the reproduction intensities $h_{i,j} = w_{i,j}m_{i,j} \to$ excitation function $\phi_{ij}$.

*Remark* G.7. The constant $\delta$ in the Theorem G.5 comes from the moment structure of the INAR sequence. For details, see Theorem 2 in (Kirchner, 2016b) and Corollary 24 in paper B. Hawkes forests in (Kirchner, 2017).

**In summary:** The linear autoregressive representation of the multivariate INAR model is established in Proposition G.3, based on the model definition provided in Definition G.1. The convergence of the multivariate INAR process to the multivariate Hawkes process, along with the correspondence of their parameters, is presented in Theorem G.5. Together, these results validate the discrete-time linear formulation stated in Theorem 4.1. This completes the proof. $\square$

## H  PROOF OF LEMMA 4.2

*Proof.* The proof of Lemma 4.2 is based on Proposition 2.2 and Theorem 2.4 from (Sullivant et al., 2010), which we restate here for completeness.

**Proposition H.1** (Rank Characterization of Conditional Independence (Sullivant et al., 2010))**.** *Let* $\mathbf{X} \sim \mathcal{N}(\mu, \Sigma)$ *be a multivariate normal random vector, and let A, B, and C be disjoint subsets of indices. Then the conditional independence statement* $\mathbf{X}_A \perp\!\!\!\perp \mathbf{X}_B \mid \mathbf{X}_C$ *holds if and only if the cross-covariance matrix* $\Sigma_{A \cup C, B \cup C}$ *has rank* $|C|$.

Although this result was originally established for linear acyclic models with independent Gaussian noise, it relies solely on second-order properties (variance and covariance) of the data and leverages path analysis rooted in the independence of noise terms. Consequently, this result remains valid for linear models with arbitrary noise distributions, since the argument applies to any distribution with finite second moments.

**Theorem H.2** (Conditional Independence in Directed Graphical Models (Sullivant et al., 2010))**.** *In a directed graph $\mathcal{G}$, a set C d-separates A and B if and only if the conditional independence statement* $\mathbf{X}_A \perp\!\!\!\perp \mathbf{X}_B \mid \mathbf{X}_C$ *holds for every distribution that is Markov with respect to $\mathcal{G}$.*

Combining the two results, we obtain the following: For any linear acyclic causal model with disjoint variable sets $\mathbf{A}_v$, $\mathbf{B}_v$, and $\mathbf{C}_v$, the set $\mathbf{C}_v$ d-separates $\mathbf{A}_v$ and $\mathbf{B}_v$ in the associated causal graph if and only if:

$$\text{rank}(\Sigma_{\mathbf{A}_v \cup \mathbf{C}_v, \mathbf{B}_v \cup \mathbf{C}_v}) = |\mathbf{C}_v|.$$

This equivalence confirms that the d-separation criterion in the causal graph corresponds to a rank condition on the cross-covariance matrix $\Sigma_{\mathbf{A}_v \cup \mathbf{C}_v, \mathbf{B}_v \cup \mathbf{C}_v}$.

The window causal graph in PO-MHP is a directed acyclic graph with linear causal relations and serially uncorrelated white noise. Therefore, the above rank condition applies directly to the window causal graph in the PO-MHP framework. This completes the proof. $\qquad\square$

## I  PROOF OF PROPOSITION 4.3

*Proof.* For any subprocess $O_1$, we prove the equivalence of the four statements step by step.

**(1) $\Leftrightarrow$ (2):** If $\mathcal{P}_\mathcal{G}$ is the parent-cause set of $O_1$ in the summary graph, by construction of the window causal graph, it equivalent to that the corresponding lagged variable set $\mathbf{P}_v$ contains all direct parent variables of $O_1^{(n)}$. This follows from the fact that, in the window graph, directed edges exist from the effective lagged variables of each parent subprocess to $O_1^{(n)}$. Moreover, by definition of the parent-cause set, $\mathcal{P}_\mathcal{G}$ is minimal with this property.

**(2) $\Leftrightarrow$ (3):** If $\mathbf{P}_v$ contains all parent variables of $O_1^{(n)}$ in the window graph, by the Markov property of DAGs, $\mathbf{P}_v$ d-separates $O_1^{(n)}$ from all other observed variables in $\mathbf{O}_v \backslash \left( \mathbf{P}_v \cup \{O_1^{(n)}\} \right)$. Reversly, if $\mathbf{P}_v$ d-separates $O_1^{(n)}$ from all other observed variables in $\mathbf{O}_v \backslash \left( \mathbf{P}_v \cup \{O_1^{(n)}\} \right)$, by the Granger causality-events in the future cannot causally influence events in the past, $\mathbf{P}_v$ should contain all parent variables of $O_1^{(n)}$ in the window graph. Moreover, by definition of the parent-cause set, $\mathcal{P}_\mathcal{G}$ is minimal with this property.

**(3) $\Leftrightarrow$ (4):** By applying Lemma 4.2, the d-separation between $O_1^{(n)}$ and the rest of the variables, conditioned on $\mathbf{P}_v$, is equivalent to the rank constraint:

$$\text{rank}\left( \Sigma_{\{O_1^{(n)}\} \cup \mathbf{P}_v, \, \mathbf{O}_v \backslash \{O_1^{(n)}\}} \right) = |\mathbf{P}_v|.$$

**(4) $\Leftrightarrow$ (1):** Assume the rank condition holds for $\mathbf{P}_v$. By Lemma 4.2, this implies that $\mathbf{P}_v$ d-separates $O_1^{(n)}$ from all other observed variables in the window graph. Translating back to the summary graph, this implies that $\mathcal{P}_\mathcal{G}$ is the minimal parent-cause set of $O_1$, as no smaller set can block all paths to $O_1$.

Thus, all statements are equivalent. This completes the proof. $\qquad\square$

## J  PRELIMINARIES FOR PROOFS OF PROPOSITION 4.5 AND THEOREMS 4.7 AND 4.8

To establish this result, we rely on the concepts of trek separation (t-separation) and d-separation introduced by (Sullivant et al., 2010), which provide powerful tools for analyzing latent structures in linear causal models.

**Definition J.1** (Trek (Sullivant et al., 2010)). A trek in the DAG $\mathcal{G}$ from variable $V_i$ to variable $V_j$ is an ordered pair of directed paths $(\mathbf{P_1}, \mathbf{P_2})$ where $\mathbf{P_1}$ has sink $V_i$, $\mathbf{P_2}$ has sink $V_j$, and both $\mathbf{P_1}$ and $\mathbf{P_2}$ have the same source $V_k$. The common source $V_k$ is called the top of the trek, denoted $\text{top}(\mathbf{P_1}, \mathbf{P_2})$. Note that one or both of $\mathbf{P_1}$ and $\mathbf{P_2}$ may consist of a single variable, that is, a path with no edges. A trek $(\mathbf{P_1}, \mathbf{P_2})$ is *simple* if the only common variable among $\mathbf{P_1}$ and $\mathbf{P_2}$ is the common source $\text{top}(\mathbf{P_1}, \mathbf{P_2})$. We let $\mathcal{T}(V_i, V_j)$ and $\mathcal{S}(V_i, V_j)$ denote the sets of all treks and all simple treks from $V_i$ to $V_j$, respectively.

**Definition J.2** (T-separation (Sullivant et al., 2010)). Let $\mathbf{A}_v$, $\mathbf{B}_v$, $\mathbf{C_A}$, and $\mathbf{C_B}$ be four subsets of total variable set $\mathbf{V}_v$. We say the ordered pair $(\mathbf{C_A}, \mathbf{C_B})$ t-separates $\mathbf{A}_v$ from $\mathbf{B}_v$ if, for every trek $(\tau_1; \tau_2)$ from a variable in $\mathbf{A}_v$ to a variable in $\mathbf{B}_v$, either $\tau_1$ contains a variable in $\mathbf{C_A}$ or $\tau_2$ contains a variable in $\mathbf{C_B}$.

**Theorem J.3** (Trek separation for directed graphical models (Sullivant et al., 2010)). *The sub-matrix $\sum_{\mathbf{A,B}}$ has rank less than equal to $r$ for all covariance matrices consistent with the graph $\mathcal{G}$ if and only if there exist subsets $\mathbf{C}_A, \mathbf{C}_B \subset \mathbf{V}_G$ with $|\mathbf{C}_A| + |\mathbf{C}_B| \leq r$ such that $(\mathbf{C}_A, \mathbf{C}_B)$ t-separates $\mathbf{A}$ from $\mathbf{B}$. Consequently,*

$$\text{rank}(\Sigma_{\mathbf{A,B}}) \leq \min\{|\mathbf{C}_A| + |\mathbf{C}_B| : (\mathbf{C}_A, \mathbf{C}_B) \text{ t-separates } \mathbf{A} \text{ from } \mathbf{B}\}$$

*and equality holds for generic covariance matrices consistent with $\mathcal{G}$.*

**Corollary J.4** (T-separation and D-separation (Sullivant et al., 2010)). *A set $\mathbf{C}$ d-separates $\mathbf{A}$ and $\mathbf{B}$ in $\mathcal{G}$ if and only if there is a partition $\mathbf{C} = \mathbf{C}_A \cup \mathbf{C}_B$ such that $(\mathbf{C}_A, \mathbf{C}_B)$ t-separates $\mathbf{A} \cup \mathbf{C}$ from $\mathbf{B} \cup \mathbf{C}$.*

Therefore, when $\mathbf{C_A}$ and $\mathbf{C_B}$ are disjoint, the combined set $\mathbf{C_A} \cup \mathbf{C_B}$ also serves as a d-separator between $\mathbf{A}$ and $\mathbf{B}$. Moreover, since the window graph in the Hawkes process is a DAG with linear relations, the above results can be directly applied after suitable adaptation to the Hawkes process setting.

# K    PROOF OF PROPOSITION 4.5

*Proof.* We prove both directions of the equivalence.

($\Leftarrow$) **If such a latent confounder $L_1$ exists, the rank condition holds.** Suppose there exists a latent confounder $L_1$ that is one common parent cause in the parent-cause set of $\{O_1, O_2\}$, and that $L_1$ together with $\{O_1, O_2\}$ makes them locally independent of other (nonempty set) subprocesses.

Given that $L_1$ and its paths to $O_1$ and $O_2$ satisfy Definition 4.4, the contribution of $L_1$ to both $O_1$ and $O_2$ in the window graph occurs through the same number of latent intermediates, resulting in an aligned contribution across time lags. In this setup, the influence of $L_1$ will appear as a shared component across the observed variables $\{O_i^{(j)}\}_{j\in\{n-m,...,n\}}^{i\in\{1,2\}}$.

Consider the window graph with $m$ considered effective lags. Following the logic of trek separation, in the window graph with $m$ effective lagged variables, the minimal choke set $\mathbf{C_A}$ that t-separates $O_1^{(n)}, O_2^{(n)}$ from the rest is given by:

$$\mathbf{C}_A := \{L_1^{(j)}\}_{j\in\{n-m,...,n-1\}} \cup \{O_i^{(n)}\}_{i\in\{1,2\}}^{j\in\{n-m,...,n-1\}}.$$

It is equivalent to that $\mathbf{C}_A$ is the minimal set that d-separates $\{O_1^{(n)}, O_2^{(n)}\}$ from the $\mathbf{O}_v \backslash \{O_1^{(n)}, O_2^{(n)}\}$.

Thus, by Theorem J.3, the generic rank of the cross-covariance matrix is bounded above by $|\mathbf{C_A}| = 2m+m = 3m$, where $2m$ comes from observed lagged variables of $\{O_1, O_2\}$ and $m$ comes from latent lagged variables of $L_1$. However, due to the structure of the excitation function $\phi_{ij}(s) = a_{ij}w(s)$, the latent subprocess $L_1$ contributes effectively as a *single* shared component across all its lagged variables, reducing the effective rank from $m$ to 1.

To explain this, we first write the structural equations for the observed variables $\{O_i^{(j)}\}_{i\in\{1,2\}}^{j\in\{n-m,...,n\}}$ as the linear regression on those check points as:

$$\begin{bmatrix} O_1^{(n)} \\ O_1^{(n-1)} \\ \cdots \\ O_1^{(n-m)} \\ O_2^{(n)} \\ O_2^{(n-1)} \\ \cdots \\ O_2^{(n-m)} \end{bmatrix} = \mathbf{E} \begin{bmatrix} L_1^{(n-1)} \\ \cdots \\ L_1^{(n-m)} \\ O_1^{(n-1)} \\ \cdots \\ O_1^{(n-m)} \\ O_2^{(n-1)} \\ \cdots \\ O_2^{(n-m)} \end{bmatrix} + \begin{bmatrix} \epsilon_{o_1}^{(n)} + \theta_{o_1}^{(0)} \\ \epsilon_{o_1}^{(n-1)} + \theta_{o_1}^{(0)} \\ \cdots \\ \epsilon_{o_1}^{(n-m)} + \theta_{o_1}^{(0)} \\ \epsilon_{o_2}^{(n)} + \theta_{o_1}^{(0)} \\ \epsilon_{o_2}^{(n-1)} + \theta_{o_1}^{(0)} \\ \cdots \\ \epsilon_{o_2}^{(n-m)} + \theta_{o_1}^{(0)} \end{bmatrix}, \tag{11}$$

$$
\mathbf{E} =
\begin{bmatrix}
\mathbf{a}_{o_1 l_1}\mathcal{I}_0^{\Delta} & \cdots & \mathbf{a}_{o_1 l_1}\mathcal{I}_0^{m\Delta} & 1 & \overset{1}{\cdots} & 1 & 0 & \overset{0}{\cdots} & 0 \\
0 & \overset{0}{\cdots} & 0 & 1 & \overset{0}{\cdots} & 0 & 0 & \overset{0}{\cdots} & 0 \\
\cdots & \cdots & \cdots & \cdots & \cdots & \cdots & \cdots & \cdots & \cdots \\
0 & \overset{0}{\cdots} & 0 & 0 & \overset{0}{\cdots} & 1 & 0 & \overset{0}{\cdots} & 0 \\
\mathbf{a}_{o_2 l_1}\mathcal{I}_0^{\Delta} & \cdots & \mathbf{a}_{o_2 l_1}\mathcal{I}_0^{m\Delta} & 0 & \overset{0}{\cdots} & 0 & 1 & \overset{1}{\cdots} & 1 \\
0 & \overset{0}{\cdots} & 0 & 0 & \overset{0}{\cdots} & 0 & 1 & \overset{0}{\cdots} & 0 \\
\cdots & \cdots & \cdots & \cdots & \cdots & \cdots & \cdots & \cdots & \cdots \\
0 & \overset{0}{\cdots} & 0 & 0 & \overset{0}{\cdots} & 0 & 0 & \overset{0}{\cdots} & 1
\end{bmatrix}
\begin{matrix} \\ \left.\rule{0pt}{26pt}\right\}m \\ \\ \\ \left.\rule{0pt}{26pt}\right\}m \\ \\ \end{matrix}
. \tag{12}
$$

where $\mathbf{a}_{o_1 l_1}$ denotes the product of the constant coefficients along the directed path, and $\mathcal{I}_o^{m\Delta}$ represents the sum of all integration combinations. For example, if $L_1 \to L_2 \to O_1$ and the intermediate latent subprocess $L_2$ has no self loop, then $\mathbf{a}_{o_1 l_1} := a_{o_1 l_2}a_{l_2 l_1}$, and $\mathcal{I}_0^{m\Delta} := \sum_{t=1}^m \left( \int_{(t-1)\Delta}^{t\Delta} w(s)ds \int_{(m-t-1)\Delta}^{(m-t)\Delta} w(s)ds \right)$, because the latent directed path is acyclic and none of the intermediate latent subprocesses involved have self-loops. In general, different length of latent paths have different $\mathcal{I}_0^{m\Delta}$, which is one reason that we require latent paths with equal lengths. However, interesting, for exponential kernel $a_{i,j}e^{(-\beta s)}$ where $w(s) := e^{(-\beta s)}$, the element $\mathcal{I}_0^{m\Delta} := \sum_{t=1}^m \left( \int_{(t-1)\Delta}^{t\Delta} w(s)ds \int_{(m-t-1)\Delta}^{(m-t)\Delta} w(s)ds \right) = \frac{-2}{\beta}(e^{-\beta m\Delta} - e^{-\beta(m-1)\Delta}) = 2\int_{(m-1)\Delta}^{m\Delta} e^{(-\beta s)}ds = 2\int_{(m-1)\Delta}^{m\Delta} w(s)ds$. Moreover, in the case of $L_1 \to L_2 \to O_1$, $\mathcal{I}_0^{\Delta}$ should be 0. This is key reason that two paths must be equal length for theoretical identifiability. Theoretically, if two paths don't have equal length, the two corresponding rows don't have equal number of leading 0s so that they can not be linearly dependent and the rank deficiency can not happen.

It is straightforward to see that the rank of the coefficient matrix $\mathbf{E}$ is $2m+1$, because the two row corresponding to $O_1^{(n)}$ and $O_2^{(n)}$ in $\mathbf{E}$ are linearly dependent (proportional to each other).

Furthermore, the cross-covariance matrix of $\{O_i^{(j)}\}_{i\in\{1,2\}}^{j\in\{n-m,\dots,n\}}$ and $\mathbf{O}_v\backslash\{O_1^{(n)}, O_2^{(n)}\}$, i.e., $\Sigma_{\{O_i^{(j)}\}_{i\in\{1,2\}}^{j\in\{n-m,\dots,n\}}, \mathbf{O}_v\backslash\{O_1^{(n)},O_2^{(n)}\}}$ can be written as $\mathbf{E}\mathbf{C}_A\mathbf{C}_A^{\top}\mathbf{F}^{\top}$ where $\mathbf{E}$ and $\mathbf{F}$ are coefficient matrix by regressing variables on those choke points. The $\text{rank}(\mathbf{C}_A\mathbf{C}_A^{\top}\mathbf{F}^{\top})$ has full column rank, because $\mathbf{F}$ calculated from regressing all the rest variables $\mathbf{O}_v\backslash\{O_1^{(n)}, O_2^{(n)}\}$ on $\mathbf{C}_A$ and without blocking lagged variables, no shrinkage of rank occurs. Consequently, the rank of the cross-covariance matrix $\text{rank}\left(\Sigma_{\{O_i^{(j)}\}_{i\in\{1,2\}}^{j\in\{n-m,\dots,n\}}, \mathbf{O}_v\backslash\{O_1^{(n)},O_2^{(n)}\}}\right) = rank\left(\mathbf{E}\mathbf{C}_A\mathbf{C}_A^{\top}\mathbf{F}^{\top}\right) = \text{rank}(\mathbf{E}) = 2m+1$ (The following theorem proofs also adopt a similar way).

Thus, the total rank becomes:

$$\text{rank} = 2m \text{ (from observed lags of } O_1 \text{ and } O_2) + 1 \text{ (from } L_1) = 2m+1.$$

**($\Rightarrow$) If the rank condition holds, there exists a latent confounder $L_1$ satisfying the claimed properties.** Conversely, assume the observed rank condition:

$$\text{rank}\left(\Sigma_{\{O_i^{(j)}\}_{i\in\{1,2\}}^{j\in\{n-m,\dots,n\}}, \mathbf{O}_v\backslash\{O_1^{(n)},O_2^{(n)}\}}\right) = 2m+1.$$

By construction of the window graph (Eq. 2), if there were no latent confounder between $O_1$ and $O_2$, the rank would be at most $2m$, corresponding to the observed lagged variables of $O_1$ and $O_2$. The observed rank being strictly $2m+1$ thus implies the presence of an additional latent variable influencing both $O_1$ and $O_2$.

Due to the rank faithfulness assumption (Assumption 2), such a rank elevation uniquely corresponds to a latent subprocess $L_1$ acting as a parent cause of both $O_1$ and $O_2$. Furthermore, for the rank increment to be exactly one, the causal paths from $L_1$ to $O_1$ and $O_2$ must satisfy the *symmetric path situation* (Definition 4.4): i.e., the paths only involve intermediate latent subprocesses of the same depth without self-loops, ensuring that the contribution of $L_1$ introduces a single additional rank component shared by both $O_1$ and $O_2$ at the same temporal lag level.

Finally, by construction, conditioning on $\mathcal{P}_{\mathcal{G}}' := L_1 \cup \{O_1, O_2\}$ removes all causal influence from $L_1$, rendering $\{O_1, O_2\}$ locally independent of the remaining observed subprocesses.

This completes the proof. $\qquad\square$

## L    PROOF OF THEOREM 4.7

*Proof.* We prove both directions of the equivalence.

($\Leftarrow$) **If such a parent-cause set $\mathcal{P}'_{\mathcal{G}}$ exists, the rank condition holds.**    Assume that $\mathcal{P}'_{\mathcal{G}}$ is the minimal set of subprocesses such that:

- $\mathcal{P}'_{\mathcal{G}}$ is a subset of the parent-cause set of $N_1$.
- Conditioning on $\mathcal{S}_{\mathcal{G}} := \mathcal{P}'_{\mathcal{G}} \cup \mathcal{D}e(N_1) \cup \{\mathcal{D}e(L_i)\}_{L_i \in \mathcal{P}'_{\mathcal{G}}} \cup \mathcal{S}ib(\mathcal{D}e(N_1)) \cup \{\mathcal{S}ib(\mathcal{D}e(L_i))\}_{L_i \in \mathcal{P}'_{\mathcal{G}}}$ renders $N_1$ locally independent of all other (nonempty set) subprocesses in the system.
- All possible observed surrogates of $N_i$ in $\mathcal{O}_{\mathcal{G}}$ have been identified.
- For each $L_i \in \mathcal{P}'_{\mathcal{G}}$, the relationship between $L_i$ and its observed effects $\{\mathcal{D}e(N_1), \mathcal{D}e(L_i)\}$ satisfies Definition 4.4.

In this setup, the lagged variables of $\mathcal{D}e(N_1)$ and $\mathcal{D}e(L_i)$, as well as the lagged and current variables of their observed siblings $\mathcal{S}ib(\mathcal{D}e(N_1))$ and $\mathcal{S}ib(\mathcal{D}e(L_i))_{L_i \in \mathcal{P}'_{\mathcal{G}}}$, appear in both $\mathbf{A}_v$ and $\mathbf{B}_v$. The rank contribution from these *observed variables* is deterministically: $|\mathbf{O}_{v1}| :=$
$$\left| \{\mathcal{D}e(N_1)^{(j)}, \mathcal{D}e(L_i)^{(j)}\}^{j \in \{n-m,\ldots,n-1\}}_{L_i \in \mathcal{P}'_{\mathcal{G}}} \cup \{O_i^{(j)}\}^{j \in \{n-m,\ldots,n-1\}}_{O_i \in \mathcal{P}'_{\mathcal{G}}} \cup \{O_i^{(j)}\}^{j \in \{n-m,\ldots,n\}}_{O_i \in \mathcal{S}ib(\mathcal{D}e(N_1)) \cup \{\mathcal{S}ib(\mathcal{D}e(L_i))\}_{L_i \in \mathcal{P}'_{\mathcal{G}}}} \right|.$$
The remaining part of $\mathbf{A}_v$, i.e., $\mathbf{A}_v \backslash \mathbf{O}_{v1}$, consists of the current variables $\{\mathcal{D}e(N_1)^{(n)}, \mathcal{D}e(L_i)^{(n)}\}_{L_i \in \mathcal{P}'_{\mathcal{G}}}$.

Given the symmetric path structure (Definition 4.4), each latent confounder $L_i \in \mathcal{P}'_{\mathcal{G}}$ contributes exactly one shared latent component, as the influence propagates through symmetric, acyclic paths. Due to the specific excitation function $\phi_{ij}(s) = a_{ij}w(s)$, this results in precisely one rank contribution per latent subprocess, regardless of the number of lagged variables.

Thus, the latent contribution adds exactly:
$$|\mathbf{O}_{v2}| := \left| \{L_i\}_{L_i \in \mathcal{P}'_{\mathcal{G}}} \right| = \left| \mathcal{D}e(L_i)^{(n)} \}_{L_i \in \mathcal{P}'_{\mathcal{G}}} \right|$$
rank-one components.

Combining both observed and latent contributions, the total rank becomes:
$$|\mathbf{O}_{v1}| + |\mathbf{O}_{v2}| = |\mathbf{O}_{v1}| + |\mathbf{O}_{v2}| + 1 \text{ (from } \mathcal{D}e(N_1)^{(n)}) - 1 = |\mathbf{A}_v| - 1.$$

($\Rightarrow$) **The rank condition implies the claimed causal structure and local independence.**    Assume that $\mathcal{P}'_{\mathcal{G}}$ is the minimal set such that:
$$\mathrm{rank}\left( \Sigma_{\mathbf{A}_v, \mathbf{B}_v} \right) = |\mathbf{A}_v| - 1$$

By the theory of trek separation (Theorem J.3), such a rank deficiency implies that the information flow between $\mathbf{A}_v$ and $\mathbf{B}_v$ must pass through a set of choke points, corresponding to the candidate parent causes in $\mathcal{P}'_{\mathcal{G}}$.

If no latent confounders existed, or if $\mathcal{P}'_{\mathcal{G}}$ were not part of the parent-cause set of $N_1$, the rank would be exactly $|\mathbf{O}_{v1}|$, solely contributed by the lagged variables of observed surrogates and both the current and lagged variables of their siblings.

Since all possible observed surrogates of $N_i$ in $\mathcal{O}_{\mathcal{G}}$ have been identified, the extra deficiency of rank (i.e., $|\mathbf{O}_{v2}|$) thus directly implies the existence of latent subprocesses contributing shared rank-one components. By the rank faithfulness, this observed rank pattern is only consistent with the existence of latent subprocesses $\{L_i\}_{L_i \in \mathcal{P}'_{\mathcal{G}}}$ that act as confounders between $\mathcal{D}e(N_1)$ and their respective observed effects, and these latent subprocesses are members of the parent-cause set of $N_1$.

For the rank deficit per latent subprocess to be exactly one, the contribution from each latent subprocess must propagate through symmetric acyclic paths, consistent with Definition 4.4, ensuring a single rank-one component contribution per latent subprocess. Moreover, the inclusion of the observed surrogates and their siblings ensures that no alternative paths can explain the dependency patterns. Thus, $\mathcal{P}'_{\mathcal{G}}$ must be the subset of parent causes, satisfying the conditional local independence of $N_1$ given $\mathcal{S}_{\mathcal{G}}$.

Therefore, the rank condition is both necessary and sufficient to identify $\mathcal{P}'_{\mathcal{G}}$ as the subset of parent causes of $N_1$, considering both observed and latent subprocesses. This completes the proof. $\qquad\square$

## M    PROOF OF THEOREM 4.8

*Proof.* We prove both directions of the equivalence.

**($\Leftarrow$) If such a latent confounder $L_1$ exists, the rank condition holds.** Assume there exists a latent confounder subprocess $L_1$ such that:

- $L_1$ is a common parent cause of $\{N_1, N_2\}$.
- Conditioning on $\mathcal{P}'_{\mathcal{G}} \coloneqq L_1 \cup N_1, N_2 \cup \mathcal{S}ib(\mathcal{D}e(N_i))_{i \in \{1,2\}}$ renders $\{N_1, N_2\}$ locally independent of the rest (nonempty set) of the system $\mathcal{N}_{\mathcal{G}} \setminus \mathcal{P}'_{\mathcal{G}}$.
- All possible observed surrogates of $\{N_1, N_2\}$ in $\mathcal{O}_{\mathcal{G}}$ have been identified.
- $L_1$ and its observed effects $\{\mathcal{D}e(N_1), \mathcal{D}e(N_2)\}$ satisfy Definition 4.4.

By the Definition 4.4, the causal influence from $L_1$ to $\{\mathcal{D}e(N_1), \mathcal{D}e(N_2)\}$ is symmetric and only propagates through the same number of intermediate latent subprocesses without self-loops. Under this condition, the contributions of $L_1$ to the observed surrogates $\{\mathcal{D}e(N_1), \mathcal{D}e(N_2)\}$ appear as a rank-one component across the lagged variables of these subprocesses, aligned in time.

Thus, in the window graph, the latent influence from $L_1$ will introduce exactly one additional rank component across the observed variable set $\mathbf{A}_v$ beyond the rank contribution from the observed lagged variables themselves.

Formally, following the arguments for Proposition 4.5, the rank of $\Sigma_{\mathbf{A}_v, \mathbf{B}_v}$ is determined by the minimal set of choke points that t-separate $\mathbf{A}_v$ from $\mathbf{B}_v$ in the window graph. Given the assumed structure:

- The lagged variables of $\{\mathcal{D}e(N_1), \mathcal{D}e(N_2)\}$ and both the current and lagged variables of their observed siblings, denoted as $\mathbf{O}_{v1} \coloneqq \{\mathcal{D}e(N_i)^{(j)}\}_{i \in \{1,2\}}^{j \in \{n-m,\dots,n-1\}} \cup \{O_i^{(j)}\}_{O_i \in \mathbf{Sib}(\mathcal{D}e(N_1)) \cup \mathbf{Sib}(\mathcal{D}e(N_2))}^{j \in \{n-m,\dots,n\}}$, appear in both $\mathbf{A}_v$ and $\mathbf{B}_v$, contributing deterministically $|\mathbf{O}_{v1}|$ to the rank.
- The influence from $L_1$ propagates symmetrically to both $\mathcal{D}e(N_1)$ and $\mathcal{D}e(N_2)$ through acyclic paths composed exclusively of latent subprocesses, per Definition 4.4. As a result, due to the excitation function $\phi_{ij}(s) = a_{ij} w(s)$, the total rank contribution from $L_1$ is exactly one.

Therefore, the total rank becomes:

$$\text{rank}\left(\Sigma_{\mathbf{A}_v, \mathbf{B}_v}\right) = |\mathbf{O}_{v1}| + 1 = |\mathbf{A}_v| - 1$$

**($\Rightarrow$) If the rank condition holds, such a latent confounder $L_1$ must exist.** Now assume the observed rank condition:

$$\text{rank}\left(\Sigma_{\mathbf{A}_v, \mathbf{B}_v}\right) = |\mathbf{A}_v| - 1$$

We know that parent-cause sets of all inferred latent confounder processes in $\mathcal{N}_{\mathcal{G}}$ remain unidentified even after applying Theorem 4.7. In the absence of any new latent confounder, the maximum possible rank would be $|\mathbf{O}_{v1}|$, corresponding solely to the contributions of the lagged variables of $\{\mathcal{D}e(N_1), \mathcal{D}e(N_2)\}$ and both the current and lagged variables of their observed siblings. The observed rank being exactly $|\mathbf{O}_{v1}| + 1 = |\mathbf{A}_v| - 1$ implies the existence of an additional latent source influencing both $N_1, N_2$ and their observed surrogates.

Due to the rank faithfulness, this increment must be attributed to a unique latent subprocess $L_1$ that acts as a confounder for $N_1$ and $N_2$. Moreover, the fact that the rank increment is only one implies that the paths from $L_1$ to $N_1, N_2$ must satisfy the symmetric and acyclic conditions in Definition 4.4, ensuring that the influence of $L_1$ is captured as a rank-one shared component at the observed surrogates level.

Moreover, the inclusion of the observed surrogates and their siblings ensures that all other possible paths and confounding structures are blocked, enforcing $\mathcal{P}'_{\mathcal{G}} \coloneqq L_1 \cup \{N_1, N_2\} \cup \{\mathcal{S}ib(\mathcal{D}e(N_i))\}_{i \in \{1,2\}}$ in ensuring local independence and all possible observed surrogates of $\{N_1, N_2\}$ in $\mathcal{O}_{\mathcal{G}}$ have been identified.

Thus, the rank pattern is both necessary and sufficient to imply the existence of $L_1$ and the claimed causal and conditional independence structure. This completes the proof. $\square$

# N  PROOF OF THEOREM 5.1

*Proof.* We prove the theorem by considering the two cases separately: (i) the system contains no latent subprocesses, and (ii) the system contains latent subprocesses.

**Case (i): No latent subprocesses.** In this case, the system consists solely of observed subprocesses $\mathcal{O}_{\mathcal{G}}$. Since there are no latent confounders, Phase I alone is sufficient for identifiability. This follows directly from Proposition 4.3, which ensures that for each observed subprocess, its parent-cause set can be uniquely identified by checking the rank condition of the relevant cross-covariance matrices. Thus, the entire causal graph can be identified solely through Phase I.

**Case (ii): Presence of latent subprocesses.** In the general case where latent subprocesses exist, the algorithm relies on the synergy between Phase I and Phase II.

- **Phase I** iteratively identifies the parent-cause set for each subprocess (including both observed and previously discovered latent subprocesses) whose parent-cause set is fully contained in the current set of known subprocesses. By Proposition 4.3 and Theorem 4.7, this identification is guaranteed when no unknown latent confounders intervene or when latent confounders are already represented by their observed surrogates.
- **Phase II** handles the discovery of new latent confounder subprocesses by systematically applying Proposition 4.5 and Theorem 4.8. The identifiability is guaranteed under the condition that all latent confounder subprocesses and their associated observed effects satisfy Definition 4.4. This condition uniquely ensures that each latent confounder subprocess contributes a unique, identifiable rank-1 pattern in the cross-covariance matrix of its observed surrogates and their siblings, enabling its detection through the rank conditions established in the theorems. The latent cofounder subprocesses that do not satisfy Definition 4.4 remain unidentified.

**Termination and completeness.** The algorithm alternates between Phase I and Phase II. Since each iteration either identifies a new parent-cause set or discovers a new latent subprocess, and given the finite number of subprocesses (including latent ones), the algorithm must eventually terminate.

By construction:

- All observed subprocesses will eventually have their parent-cause sets identified through Phase I.
- All latent subprocesses satisfying Definition 4.4 will be identified through Phase II and incorporated into the active set for further investigation.
- The recursive application of the identification theorems ensures that no causal relationships (either between observed, latent, or between observed and latent) will remain unidentified under the conditions.
- If Definition 4.4 fails for any latent, the algorithm terminates without fabricating that latent or any edges it would entail, thereby returning only the identifiable portion of the causal graph (sound abstention).

Thus, under excitation function $\phi_{ij}(s) = a_{ij}w(s)$ and rank faithfulness, the entire causal graph consisting of both observed subprocesses and latent confounders can be identified. This completes the proof. $\square$

## O    Details of identification algorithm

### O.1    Phase I

The detailed algorithm for Phase I is in Algorithm 2.

---

**Algorithm 2** Identifying Causal Relations

---

**Input:** Partial causal graph $\mathcal{G}$, Active subprocess set $\mathcal{A}_{\mathcal{G}}$, Observed subprocess set $\mathcal{O}_{\mathcal{G}}$
**Output:** Partial causal graph $\mathcal{G}$, Active subprocess set $\mathcal{A}_{\mathcal{G}}$

1: **repeat**
2:     Select a subprocess $N_1$ from $\mathcal{A}_{\mathcal{G}}$.
3:     **for** $Len = 1$ **to** $|\mathcal{A}_{\mathcal{G}} \cup \mathcal{O}_{\mathcal{G}}|$ **do**
4:         **repeat**
5:             Select subset $\mathcal{P}'_{\mathcal{G}} \subseteq \mathcal{A}_{\mathcal{G}} \cup \mathcal{O}_{\mathcal{G}}$ such that $|\mathcal{P}'_{\mathcal{G}}| = Len$.
6:             **if** $(\mathcal{A}_{\mathcal{G}} \cup \mathcal{O}_{\mathcal{G}}, \mathcal{P}'_{\mathcal{G}}, N_1)$ satisfies Proposition 4.3 and Theorem 4.7 **then**
7:                 $\mathcal{A}_{\mathcal{G}} = \mathcal{A}_{\mathcal{G}} \backslash N_1$, and update $\mathcal{G}$.
8:                 Return to line 2.
9:             **end if**
10:         **until** All subsets of $\mathcal{A}_{\mathcal{G}} \cup \mathcal{O}_{\mathcal{G}}$ with size $Len$ selected.
11:     **end for**
12: **until** $\mathcal{A}_{\mathcal{G}}$ is not updated or $|\mathcal{A}_{\mathcal{G}}| \leq 1$.
13: **return:** $\mathcal{G}$, $\mathcal{A}_{\mathcal{G}}$

---

### O.2    Phase II

The detailed algorithm for Phase II is in Algorithm 3.

## P    Computational Complexity of the Algorithm

In this section, we analyze the computational complexity of our two-phase iterative algorithm, which alternates between: (1) inferring causal relationships among discovered subprocesses and (2) identifying new latent subprocesses. Let $n$ denote the number of processes in the *active process set* $\mathcal{A}_{\mathcal{G}}$ and $m$ denote the number of

---

**Algorithm 3** DiscoveringNewLatentComponentProcesses

---

**Input:** Partial causal graph $\mathcal{G}$, Active subprocess set $\mathcal{A}_{\mathcal{G}}$, Observed subprocess set $\mathcal{O}_{\mathcal{G}}$
**Output:** Partial causal graph $\mathcal{G}$, Active subprocess set $\mathcal{A}_{\mathcal{G}}$

1: Initialize cluster set $\mathbb{C} := \emptyset$ and the group size $Len = 2$.
2: **repeat**
3:     Select a subset $\mathcal{Y}_{\mathcal{G}}$ from $\mathcal{A}_{\mathcal{G}}$ such that $|\mathcal{Y}_{\mathcal{G}}| = Len$.
4:     **if** $(\mathcal{A}_{\mathcal{G}} \cup \mathcal{O}_{\mathcal{G}}, \mathcal{Y}_{\mathcal{G}})$ satisfies Proposition 4.5 and Theorem 4.8 **then**
5:         Add $\mathcal{Y}_{\mathcal{G}}$ into $\mathbb{C}$.
6:     **end if**
7: **until** All subset of $\mathcal{A}_{\mathcal{G}}$ with size $Len$ selected.
8: Merge all the overlapping sets in $\mathbb{C}$.
9: **for** each merged set $\mathcal{C}_i \in \mathbb{C}$ **do**
10:     Introduce a new latent subprocess $L_j$.
11:     $\mathcal{A}_{\mathcal{G}} = \mathcal{A}_{\mathcal{G}} \cup L_j \backslash \mathcal{C}_i$, and update $\mathcal{G}$.
12: **end for**
13: **return:** $\mathcal{G}, \mathcal{A}_{\mathcal{G}}$

---

subprocesses in the augmented process set $\mathcal{T}_{\mathcal{G}} := \mathcal{A}_{\mathcal{G}} \cup \mathcal{O}_{\mathcal{G}}$ at the start of each phase. Assume each test is an oracle test.

**Phase I: Inferring Causal Relationships**

For each component process $N_1 \in \mathcal{A}_{\mathcal{G}}$, we evaluate subsets of $\mathcal{T}_{\mathcal{G}}$ starting from subsets of size 1 up to the size of $\mathcal{T}_{\mathcal{G}}$, stopping when the test result is positive. In the worst case, for each $N_1$, we need to evaluate all subsets of $\mathcal{T}_{\mathcal{G}}$, which requires $\sum_{k=1}^{m} \binom{m}{k}$ tests. For one subprocess $N_1 \in \mathcal{A}_{\mathcal{G}}$, if its parent-cause set is found, $\mathcal{A}_{\mathcal{G}}$ is updated. After that, the algorithm will restart to go over all the subprocesses in $\mathcal{A}_{\mathcal{G}}$ to make sure no parent-cause set of subprocesses in $\mathcal{A}_{\mathcal{G}}$ can be found. In the worst case, the algorithm find parent-cause set for the last component process in $\mathcal{A}_{\mathcal{G}}$ each time. The complexity of Phase I is upper bounded by: $\mathcal{O}\left(n! \sum_{k=1}^{m} \binom{m}{k}\right)$.

**Phase II: Identifying New Latent Subprocesses**

In this phase, we test all subsets of $\mathcal{A}_{\mathcal{G}}$ of size 2. Since there are $\binom{n}{2}$ such subsets, the complexity of Phase II is upper bounded by: $\mathcal{O}\left(\binom{n}{2}\right)$.

**Overall Complexity**

The total complexity of the algorithm depends on the number of (both observed and latent) subprocesses and the structural density of the causal graph, as these factors determine the number of iterations required for the algorithm to run. Combining the two phases, for each iteration, the overall complexity is approximately upper bounded by: $\mathcal{O}\left(n! \sum_{k=1}^{m} \binom{m}{k} + \binom{n}{2}\right)$.

In practical scenarios, the structural density of the causal graph and sparsity of dependencies may reduce the number of required iterations and tests, leading to improved efficiency compared to this worst-case analysis.

## Q    MORE DETAILS OF EXPERIMENTS

### Q.1    SYNTHETIC DATA GENERATION AND IMPLEMENTATION

We evaluate our method on two types of synthetic data: event sequences generated by the Hawkes process in Eq. (1), and discrete-time data generated directly from the discrete-time model in Eq. (2)

**Hawkes Process Data**: We generate event sequences using the `tick` library (Bacry et al., 2017), an efficient framework for simulating multivariate Hawkes processes. The excitation function is set as exponential kernel $\phi_{ij}(s) = \alpha_{ij} e^{-\beta s}$, where $\beta$ is fixed at 1. $\alpha_{ij}$ for edges involving latent subprocesses are sampled uniformly from $[0.90, 0.99]$, and $\alpha_{ij}$ for other edges including observed self-loops are sampled uniformly from $[0.30, 0.50]$, because the causal influences attenuate along latent paths. And in Case 1, sampling $\alpha_{ij}$ from $[0.30, 0.50]$ avoids the cycle between $N_2$ and $N_3$ becoming nonstationary. Moreover, to ensure stationarity and avoid explosive behavior, we verify the spectral radius of the integrated excitation matrix after generating $\alpha_{ij}$. To discretize the sequences for our method, we select the time bins of length 0.1 and consider 600 effective lag time bins as discretized lagged variables for the calculation sub-covariance matrix. The sample size corresponds to the number of discrete data points.

**Discrete-Time Series Data**: To assess our method under ideal discrete-time conditions (i.e., exactly satisfying Theorem 4.1), we generate data directly from Eq. (2). The excitation function is set as exponential kernel

$\phi_{ij}(s) = \alpha_{ij}e^{-\beta s}$. The coefficients $\alpha_{ij}$ and decay parameter $\beta$ are set as above. Similar to the Hawkes data, we verify the spectral radius to ensure stationarity. The noise terms are drawn from independent Gaussian distributions. We set the number of effective lagged variables to 200. The sample size corresponds to the number of discrete data points.

**Preprocessing and Rank Deficiency Testing**: For each trial, we standardize the discretized data to ensure fair comparison. To test for rank deficiency, we use canonical correlation analysis (CCA) (w Anderson, 1974), following the procedure in (Huang et al., 2022). We use the grid search to find the best rank test threshold. We also conduct a empirical sensitivity analysis for test threshold. The result is in Appendix Q.3.

**Data Usage for Baselines**: For Hawkes process-based methods (SHP (Qiao et al., 2023), THP (Cai et al., 2022), and NPHC (Achab et al., 2018)), we use the raw Hawkes process data produced by the `tick` library. For rank-based methods designed for i.i.d. data with linear relations (Hier. Rank (Huang et al., 2022) and RLCD (Dong et al., 2023)) and time series baseline (LPCMCI (Gerhardus & Runge, 2020), we use the discretized Hawkes process data.

We run all the experiments on a personal CPU machine.

## Q.2 EVALUATION METRICS

We evaluate the accuracy of causal structure recovery using the standard F1-score, which combines precision and recall.

Causal relationships among both latent and observed subprocesses are represented by an adjacency matrix, where each entry is either 1 or 0, indicating the presence or absence of a directed edge, respectively. Specifically, $\text{Adj}_{\mathcal{G}}(i, j) = 1$ denotes a directed edge from the $j$-th subprocess to the $i$-th subprocess, while $\text{Adj}_{\mathcal{G}}(i, j) = 0$ indicates no such edge.

We measure the similarity between the estimated and ground-truth adjacency matrices using the F1-score. First, we compute precision, defined as

$$\text{precision} = \frac{\text{true positives}}{\text{total inferred positives}},$$

which represents the proportion of correctly inferred edges among all predicted edges. Next, we calculate recall, defined as

$$\text{recall} = \frac{\text{true positives}}{\text{total ground-truth positives}},$$

which captures the proportion of correctly inferred edges relative to the true causal edges. The F1-score, given by

$$\text{F1-score} = 2 \cdot \frac{\text{precision} \times \text{recall}}{\text{precision} + \text{recall}},$$

harmonizes precision and recall to provide a balanced measure of structural recovery.

**Practical Considerations**

In practice, the indices of latent subprocesses in the estimated (summary) graph may not correspond to those in the ground truth. To address this, following Huang et al. Huang et al. (2022), we permute and divide the latent subprocess indices in the estimated graph and select the result that minimizes the difference from the true graph. When the number of estimated latent subprocesses is smaller than the true number after division, we add isolated latent nodes to balance the comparison. Conversely, if the estimate exceeds the true number, we merge and select the subset that best matches the true latent subprocesses.

Additionally, since our inferred summary graph simplifies the underlying causal structure, by omitting intermediate latent subprocesses and redundant edges as formalized in our theorems and Definition 4.4, we adjust the ground-truth adjacency matrix to this idealized representation before comparison. This ensures a fair evaluation of causal discovery.

For baselines designed for i.i.d. data with linear relations (i.e., Hier. Rank (Huang et al., 2022) and RLCD (Dong et al., 2023)), their output graphs capture relationships among discretized variables, rather than subprocesses. To enable fair comparison, we regard an edge $N_1 \rightarrow N_2$ as correctly identified if more than half of the considered variables associated with $N_1$ have inferred edges to those of $N_2$.

## Q.3 ADDITIONAL SYNTHETIC EXPERIMENT RESULTS

**Comparisons on Cases 5 and 6** Fig. 8 shows the F1-score comparisons for Cases 5 and 6, which correspond to intricate latent confounder structures illustrated in Fig.3c and Fig.3d. These cases involve interactions

between latent confounders. The results indicate that our method maintains strong performance even under these challenging causal configurations.

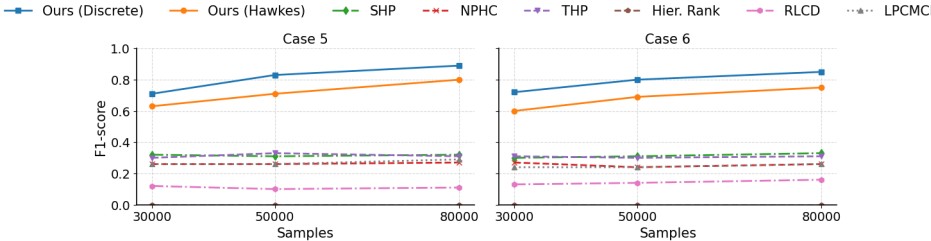

Figure 8: F1-score comparisons on the remaining two causal graphs (Cases 5 and 6), involving latent confounder interactions. Case 5 and Case 6 correspond to the causal structures in Figs. 3c and 3d, respectively.

**Sensitivity to Time Discretization Interval**     We evaluate the sensitivity of our method to the choice of the discretization interval $\Delta$ with decay parameter $\beta = 1$ in the exponential excitation function $\phi_{ij}(s) = \alpha_{ij} e^{-\beta s}$. As shown in Table 1, when $\Delta$ is set to 0.01 or 0.05, our method achieves consistently high F1-scores across all cases, confirming that the discretized representation sufficiently preserves the temporal dynamics of the underlying Hawkes process. Even at $\Delta = 0.1$, the performance remains stable. However, when $\Delta$ increases to 0.3, we observe a sharp drop in performance, highlighting that overly coarse discretization leads to significant loss of temporal resolution, impairing the estimation of causal structures. The result shows the need to choose a small bin width $\Delta$ relative to the typical support of the excitation function (Kirchner, 2016a).

Table 1: Performance of our method under varying $\Delta$ values using 80k Hawkes process samples generated by the `tick` library with decay parameter $\beta = 1$ in the exponential excitation function. Case 1–3 correspond to Figs. 1b, 2a, and 3a, respectively. Results are averaged over ten runs. Performance remains stable and high when $\Delta \leq 0.1$, but degrades significantly at $\Delta = 0.3$ due to the loss of fine-grained temporal information.

| $\Delta$ | **Precision** | | | **Recall** | | | **F1-Score** | | |
|---|---|---|---|---|---|---|---|---|---|
|  | Case 1 | Case 2 | Case 3 | Case 1 | Case 2 | Case 3 | Case 1 | Case 2 | Case 3 |
| 0.01 | 0.98 | 0.91 | 0.84 | 0.92 | 0.93 | 0.83 | 0.93 | 0.92 | 0.84 |
| 0.05 | 1.00 | 0.96 | 0.83 | 0.84 | 0.98 | 0.82 | 0.90 | 0.97 | 0.82 |
| 0.10 | 1.00 | 0.91 | 0.86 | 0.87 | 0.93 | 0.83 | 0.93 | 0.92 | 0.84 |
| 0.30 | 0.50 | 0.55 | 0.50 | 0.17 | 0.63 | 0.33 | 0.25 | 0.59 | 0.40 |

**Sensitivity to Rank-Test Threshold**     We evaluate the sensitivity of our method to the threshold $\tau$ used in the rank test (i.e., the cutoff deciding rank deficiency). We vary $\tau \in \{0.01, 0.05, 0.10, 0.20\}$ and assess three representative cases. Each experiment uses 30k Hawkes samples generated by the `tick` library under an exponential excitation function $\phi_{ij}(s) = \alpha_{ij} e^{-\beta s}$ with $\beta = 1$ and time interval $\Delta = 0.1$; results are averaged over ten runs. As shown in Table 2, in the fully observed setting (Case 1) precision remains 1.00 while recall decreases as $\tau$ increases, whereas in latent settings (Cases 2–3) a moderately larger threshold improves precision because of the attenuation of causal influences through the latent subprocesses. Overall, a threshold of 0.10 provides a good balance across different scenarios.

**Robustness to Violations of Rank Faithfulness**     To test robustness under violations of rank faithfulness, we randomly select two edges in each synthetic graph and assign them identical coefficients $\alpha_{ij}$ for the exponential excitation function $\phi_{ij}(s) = \alpha_{ij} e^{-\beta s}$ in every run, with $\beta = 1$ and time interval $\Delta = 0.1$. This manipulation introduces potential linear dependencies in the cross-covariance matrix, which could challenge rank-based methods. As presented in Table 3, despite the induced degeneracy, our method maintains strong performance, especially as the sample size increases. These results suggest that our approach is robust to moderate violations of rank faithfulness in practical scenarios.

**Sensitivity to smaller sample size**     We also examine robustness under reduced data availability by evaluating all methods using only 5,000 samples on Case 1 (Fig. 1b), where the underlying graph is fully observed

Table 2: Sensitivity to the rank-test threshold $\tau$. Each entry is averaged over ten runs on 30k samples generated with an exponential kernel ($\beta = 1$). Case 1–3 correspond to Figs. 1b, 2a, and 3a, respectively. Overall, a threshold of 0.10 provides a good balance across different scenarios.

| Threshold $\tau$ | Precision | | | Recall | | | F1-Score | | |
|---|---|---|---|---|---|---|---|---|---|
| | Case 1 | Case 2 | Case 3 | Case 1 | Case 2 | Case 3 | Case 1 | Case 2 | Case 3 |
| 0.01 | 1.00 | 0.42 | 0.57 | 0.80 | 0.53 | 0.50 | 0.88 | 0.47 | 0.53 |
| 0.05 | 1.00 | 0.62 | 0.62 | 0.64 | 0.73 | 0.54 | 0.77 | 0.67 | 0.57 |
| 0.10 | 1.00 | 0.66 | 0.72 | 0.60 | 0.75 | 0.65 | 0.74 | 0.71 | 0.68 |
| 0.20 | 1.00 | 0.76 | 0.68 | 0.47 | 0.85 | 0.63 | 0.62 | 0.80 | 0.65 |

Table 3: Performance of our method when, in each run, two edges in each graph are randomly assigned identical coefficients $\alpha_{ij}$ for the exponential excitation function, increasing the risk of rank deficiency. Hawkes process samples are generated by the `tick` library. Case 1–3 correspond to Figs. 1b, 2a, and 3a, respectively. Results are averaged over ten runs. Despite these perturbations, our method maintains strong performance, demonstrating robustness to such violations.

| #Samples | Precision | | | Recall | | | F1-Score | | |
|---|---|---|---|---|---|---|---|---|---|
| | Case 1 | Case 2 | Case 3 | Case 1 | Case 2 | Case 3 | Case 1 | Case 2 | Case 3 |
| 30k | 0.87 | 0.60 | 0.72 | 0.87 | 0.75 | 0.71 | 0.87 | 0.67 | 0.71 |
| 50k | 0.92 | 0.83 | 0.76 | 0.84 | 0.82 | 0.73 | 0.87 | 0.82 | 0.74 |
| 80k | 0.95 | 0.84 | 0.83 | 0.90 | 0.83 | 0.80 | 0.92 | 0.83 | 0.81 |

and contains no latent confounders. All baseline methods, except Hier.Rank, which assumes no direct edges among observed subprocesses, were included. The results (mean ± standard deviation) in Table 4 show that both variants of our method maintain competitive Precision and F1-scores under this low-sample case. These findings indicate that our approach remains comparatively robust even when only limited observational data are available.

Table 4: Performance of all methods on Case 1 (fully observed, no latent confounders) using only 5k samples. Values are mean scores across ten runs, with standard deviation shown in parentheses.

| | Ours (Discrete) | Ours (Hawkes) | SHP | NPHC | THP | RLCD | LPCMCI |
|---|---|---|---|---|---|---|---|
| Precision | 0.95 (0.10) | 0.97 (0.10) | 1.00 (0.00) | 0.53 (0.27) | 0.68 (0.27) | 0.40 (0.36) | 0.92 (0.13) |
| Recall | 0.62 (0.13) | 0.57 (0.13) | 0.93 (0.13) | 0.36 (0.10) | 0.56 (0.30) | 0.13 (0.10) | 0.53 (0.10) |
| F1-score | 0.74 (0.11) | 0.71 (0.12) | 0.96 (0.08) | 0.42 (0.09) | 0.60 (0.28) | 0.19 (0.13) | 0.67 (0.10) |

**Sensitivity to violations of the symmetric path condition** We further evaluate the robustness of the proposed method when the symmetric acyclic path condition in Definition 4.4 is violated. Starting from Case 2 in Fig. 2a, we modify the path between $L_1$ and $O_1$ by introducing additional intermediate latent subprocesses: (i) inserting one latent variable (Case A), and (ii) inserting two latent variables (Case B). These modifications make the path from $L_1$ to $O_1$ longer than the paths to the remaining observed subprocesses, thereby breaking the symmetry required for rank deficiency. Table 5 reports the resulting precision, recall, and F1-scores (mean ± standard deviation). While such violations theoretically prevent $O_1$ from being identifiable as a child of $L_1$, the empirical performance shows only mild degradation. This is partly because exponential kernels preserve proportionality across products of segmented integrals, and small differences in leading zeros induced by path length differences may be statistically weak in finite samples. These results suggest that the proposed method can remain practically robust under mild deviations from the symmetric path structure.

**Evaluation on a Larger and More Complex Causal Graph** We further evaluate our method on a larger causal graph with 14 subprocesses, as shown in Fig. 9. Table 6 reports the F1-scores averaged over ten runs. Despite the increased complexity, our method successfully recovers the underlying causal structure with high accuracy.

**Scalability and Runtime Profiling** We profile runtime on three representative synthetic graphs (Cases 1–3) and two real-world settings. All runs were executed on an AMD EPYC 9454 CPU. The first real-world

Table 5: Performance under violations of the symmetric path condition in Definition 4.4 (Case 2 in Fig. 2a). Values are mean $\pm$ standard deviation.

|  | Precision | Recall | F1-score |
|---|---|---|---|
| Case A | 0.91 (0.14) | 0.93 (0.11) | 0.92 (0.13) |
| Case B | 0.89 (0.15) | 0.90 (0.12) | 0.89 (0.13) |

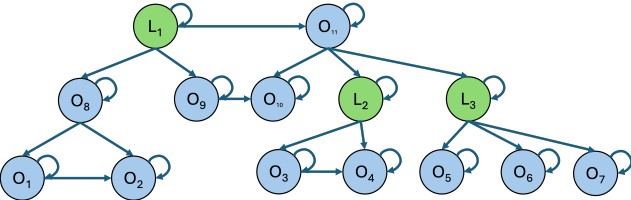

Figure 9: Illustration of a larger causal graph consisting of 14 subprocesses, used to evaluate scalability and robustness.

setting follows our main paper: a five-alarm subgraph (`Alarm_ids=0-3` with one latent `Alarm_id=7`) from *device_id* = 8. The second merges all devices into a single multivariate event sequence with all 18 alarms to gauge scaling with graph size. We observe that Case 1 is fastest as no latent confounders are present and Phase I suffices. Case 2 introduces latent confounders, requiring both phases in the first iteration and increasing runtime. Case 3 is slowest among synthetic cases because the latent confounder is itself caused by an observed subprocess, triggering an additional iteration to identify its observed parent. For real data, merging all devices markedly increases runtime as the sequence spans all 18 alarms and may deviate from a homogeneous Hawkes mechanism. A phase-wise complexity breakdown is provided in Appendix P, which offers further insight into the scalability of the algorithm.

### Q.4 ANALYSIS OF REAL-WORLD DATASET RESULTS

We evaluate our method on a real-world cellular network dataset (Qiao et al., 2023), which includes expert-validated ground-truth causal relationships. The dataset comprises 18 distinct alarm types and $\sim$35,000 recorded alarm events collected over eight months from an operational telecommunication network. This benchmark has been widely used in prior work (e.g., the PCIC 2021 causal discovery track and (Qiao et al., 2023)), where performance for many methods is available and top F1-scores are reported up to $\approx$ 0.6.

For our evaluation, we focus on a subgraph involving five alarm types (`Alarm_ids=0-3` and 7) from *device_id* = 8, where `Alarm_id=7` is manually excluded and treated as a latent subprocess. Both `Alarm_id=1` and `Alarm_id=3` are observed effects of this latent subprocess, providing an opportunity to assess our method's ability to infer latent confounders. The ground-truth causal subgraph is shown in Figure 10. Compared with our inferred causal graph, the ground truth contains an additional edge from `Alarm_id=1` to `Alarm_id=3`. However, as noted in Definition 4.4, causal edges between observed effects of a latent confounder are permissible in our framework.

During inference, using Proposition 4.3 and Theorem 4.7, we correctly identify `Alarm_ids=0,1,3` as the parent causes of `Alarm_id=2`, and `Alarm_ids=1,3` as the parent causes of `Alarm_id=0`. The parent-cause sets of `Alarm_id=1` and `Alarm_id=3` cannot be fully explained by the observed subprocesses alone. This necessitates the existence of a latent confounder influencing both, leading to the successful identification of `Alarm_id=7` as a latent subprocess.

Table 6: Performance of our method on the larger causal graph in Fig. 9, using Hawkes process data generated by the `tick` library. Results are averaged over ten runs. The method consistently recovers the causal structure with improving accuracy as sample size increases.

| Sample Size | Precision | Recall | F1-score |
|---|---|---|---|
| 30k | 0.65 | 0.52 | 0.58 |
| 50k | 0.71 | 0.58 | 0.64 |
| 80k | 0.80 | 0.71 | 0.75 |

Table 7: Runtime across synthetic and real-world settings.

| Graph Type | Runtime (s) |
|---|---|
| Case 1 | 227.80 |
| Case 2 | 1036.01 |
| Case 3 | 2603.95 |
| Real Dataset (`Alarm_ids=0--3`, *device_id = 8*) | 1364.71 |
| Real Dataset (all devices merged; 18 alarms) | 20914.29 |

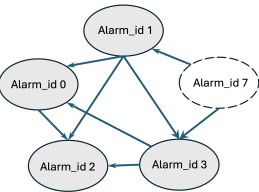

Figure 10: Ground-truth causal subgraph from the metropolitan cellular network dataset. `Alarm_id=7` is treated as a latent subprocess.

**Baselines and Schemes.** We also quantitatively compare our method with representative Hawkes-based approaches (SHP (Qiao et al., 2023), THP (Cai et al., 2022), NPHC (Achab et al., 2018)), two rank-based latent-variable methods for i.i.d. data (Hier. Rank (Huang et al., 2022), RLCD (Dong et al., 2023)), and a time-series method designed for exogenous latents (LPCMCI (Gerhardus & Runge, 2020)). For fairness, all baselines are evaluated on the same sub-dataset (`Alarm_ids=0--3` and 7 from *device_id=8*) as our method, with performance averaged over ten runs using parameter grid search.

**Results on the Sub-dataset.** Our method achieves the best F1-score when the data conforms to a single multivariate Hawkes process (per-device setting). Table 8 reports the average F1-scores.

Table 8: F1-scores on the cellular network sub-dataset (`Alarm_ids=0-3` and 7, *device_id = 8*) where `Alarm_id=7` is manually excluded and treated as a latent subprocess; averages over 10 runs.

| Algorithm | F1-score |
|---|---|
| SHP | 0.49 |
| THP | 0.48 |
| NPHC | 0.42 |
| Hier. Rank | 0.00 |
| RLCD | 0.39 |
| LPCMCI | 0.43 |
| **Ours** | **0.76** |

**Merged-devices analysis.** For completeness, we also merge events from all 55 devices into a single multivariate sequence with all 18 alarm types and analyze it with our method. This setting violates the assumption that samples share the same generative mechanism (devices can be heterogeneous), and it yields a much lower F1-score (0.17). This illustrates why per-device analysis is more compatible with our assumptions, whereas merged-device data can confound structure learning.

**Dataset description.** The dataset records 34,838 alarm events from a metropolitan cellular network (Qiao et al., 2023), covering 18 alarm types and 55 devices. Each record contains:

- **Alarm ID**: one of 18 alarm types,
- **Device ID**: one of 55 devices,
- **Start Timestamp**: time when the alarm was triggered,
- **End Timestamp**: time when the alarm was resolved.

For causal analysis, we sort events by alarm type and use the start timestamp as the event time, yielding a temporally ordered sequence suitable for inference.

