# OpenReview forum: "Causal Structure Learning in Hawkes Processes with Complex Latent Confounder Networks"
_ICLR.cc/2026/Conference — ICLR 2026 Oral_

### Official Review · Reviewer_VH6W · 2025-10-30

**Soundness:** 3
**Presentation:** 3
**Contribution:** 3
**Rating:** 8
**Confidence:** 3

**Summary:**

The paper proposes a causal discovery method for identifying latent subprocesses and their causal structures in partially observed multivariate Hawkes processes. The authors introduce a causal model for partially observed multivariate Hawkes processes to represent continuous-time event sequences. Based on this model, they leverage rank constraints on the covariance matrix to identify causal influences without prior knowledge of the existence or number of latent subprocesses.

**Strengths:**

1. The authors discretize the Hawkes process, transforming the multivariate Hawkes process causal model into a linear autoregressive model, and theoretically prove this conclusion.
2. It proposed a method for identifying latent subprocesses and causal structures solely by leveraging rank constraints on second-order statistics.

**Weaknesses:**

1. In Proposition 4.5, should it be that the rank constraint is both necessary and sufficient for the corresponding local independence in the graph, under the structure defined in Definition 4.4 and the data generated accordingly?
2. The results mentioned in the original SHP paper differ significantly from those presented in this paper. Additionally, could results for different time intervals be provided, similar to the approach in the SHP paper?
3. When the paper transforms the model into a linear autoregressive model, does it require the noise to be constrained as Gaussian?
4. The proposed two-phase iterative algorithm suffers from severe scalability limitations, which restrict its practical applicability.


Typo:

There is an extra } at line 300.

**Questions:**

What parameters were used for the method proposed in the real-world data experiments?

---

> ### Author Response · Authors · 2025-11-21
> **Rebuttal by Authors (Part 1/2)**
>
> We thank the reviewer for the insightful comments and valuable feedback. Please see our responses to your concerns point-by-point below.
>
> > **Q1: In Proposition 4.5, should it be that the rank constraint is both necessary and sufficient for the corresponding local independence in the graph, under the structure defined in Definition 4.4 and the data generated accordingly?**
>
> **A1:** Thank you for your insightful comment. **We would like to clarify that Proposition 4.5 does not assume that the underlying latent subprocesses must satisfy the symmetric path situation in Definition 4.4. Instead, Definition 4.4 provides a precise characterization of the graphical configurations with which rank deficiency arises in the relevant sub–covariance matrices of the observed subprocesses.**
>
> As shown in Appendix K (Eqs. (11)–(12)), under the excitation form $\phi_{ij}(s)=a_{ij}w(s)$ and rank faithfulness, the coefficient matrix becomes rank-deficient (i.e., linear dependence among rows of the coefficient matrix) if and only if the influences of the latent subprocess $L_1$ to both $O_1$ and $O_2$ traverse the same number of intermediate latent subprocesses. **This symmetry aligns the temporal contribution patterns across lags and produces the linear dependence that underlies the rank constraint.**
>
> **Whenever this symmetry is violated, such as through unequal path lengths or additional cross-links, the temporal contributions no longer align, and the corresponding matrix remains full rank.** Thus, the rank constraint in Proposition 4.5 is both necessary and sufficient for the local independence encoded by the path conditions in Definition 4.4.
>
> > **Q2: The results mentioned in the original SHP paper differ significantly from those presented in this paper. Additionally, could results for different time intervals be provided, similar to the approach in the SHP paper? **
>
> **A2:** Thank you for the question. The discrepancy between the SHP results and ours arises because the two methods are evaluated under fundamentally different settings. **SHP is designed for causally sufficient Hawkes processes, that is, systems without latent subprocesses. In contrast, several of our experimental cases (Cases 2–4 in Fig. 4) include latent confounder subprocesses, for which SHP is not designed and therefore exhibits degraded performance.**
>
> This is consistent with the original SHP paper: **when no latent subprocesses are present, SHP performs well. In our experiments, this corresponds to Case 1 of Fig. 4,** where SHP again achieves high F1-scores. The performance gap in the other cases reflects the model mismatch rather than an inconsistency with the SHP results.
>
> Regarding evaluation across different time intervals, **we provide such sensitivity analysis in Appendix Q.3, where we vary the discretization interval $\Delta$ and report Precision, Recall, and F1-score.** This analysis plays a similar role to the multi-interval evaluation presented in the SHP paper, and it demonstrates that our method remains stable across a range of interval choices.
>
> > **Q3: When the paper transforms the model into a linear autoregressive model, does it require the noise to be constrained as Gaussian?**
>
> **A3:** Thank you for this insightful comment. **The linear autoregressive representation does not require the noise to be Gaussian.** As established in Theorem 4.1 (proved via Proposition G.3), as the discretization interval shrinks, the Hawkes process has an equivalent linear causal representation in which the noise term $\epsilon_i^{(n)}$ forms a serially uncorrelated white-noise sequence. The argument relies only on uncorrelatedness across time; no distributional assumption is imposed. Gaussian white noise is merely a special case, not a requirement for the representation or for any identifiability result in the paper.

---

> ### Author Response · Authors · 2025-11-21
> **Rebuttal by Authors (Part 2/2)**
>
> > **Q4: The proposed two-phase iterative algorithm suffers from severe scalability limitations, which restrict its practical applicability.**
>
> **A4:** Thank you for your thoughtful consideration. The primary goal of this paper is to establish the theoretical identifiability of latent structures in Hawkes processes and to provide a constructive algorithm that demonstrates the feasibility of recovering such structures in practice. **As is common in works focusing on identifiability, the proposed algorithm serves as a proof-of-concept implementation of the theory rather than a final, fully optimized scalable solution.**
>
> Scalability limitations are shared by many causal discovery methods. Our framework provides the first characterization of identifiability for latent subprocesses in Hawkes systems through testable second-order statistics; developing more efficient algorithms that exploit this structure is a natural next step. We have outlined this direction for future work in the revised paper.
>
> > **Q5: What parameters were used for the method proposed in the real-world data experiments?**
>
> **A5:** Thank you for the question. In the real-world experiment, we used a binning interval of 1000 and a rank-test threshold of 0.1 for our method.
>
> > **Q6: Typo: There is an extra } at line 300.**
>
> **A6:** Thank you for spotting out this typo and we have now fixed it.
>
> Thank you again for your valuable feedback and hope that all your concerns are properly addressed.

---

### Official Review · Reviewer_gH4E · 2025-10-31

**Soundness:** 4
**Presentation:** 4
**Contribution:** 3
**Rating:** 8
**Confidence:** 4

**Summary:**

This paper addresses the problem of causal structure discovery in multivariate Hawkes processes under partial observability. Authors consider Hawkes processes with elaborate latent structure, and derive the conditions under which the latent variables and the relationship between them is identifiable. Similar results exist in the context of linear autoregressive processes and as acknowledged by the authors, they inspire the latent structure discovery results in this manuscript (Theorem 4.7 and 4.8). By transforming the Hawkes process inference problem into a discrete-time linear autoregressive formulation (Theorem 4.1), the authors establish the results for Hawkes processes.

The paper proposes a two-phase iterative algorithm that alternates between (i) discovering causal relations among existing subprocesses and (ii) inferring new latent subprocesses based on rank constraints of cross-covariance matrices. Necessary and sufficient conditions are derived for identifiability, including the introduction of path-based conditions (Definition 3.4) ensuring one-to-one correspondence between latent confounder structures and observable rank deficiencies. Empirical results on synthetic and real-world data show that the proposed method successfully recovers causal structures even when latent confounders exist.

**Strengths:**

Compared to the previous NeurIPS 2025 submission that I had reviewed, the manuscript has substantially improved in clarity of the claims and structure of the paper. Most importantly, the iterative algorithm is now explicitly defined, assumptions and identifiability conditions are more carefully motivated, and the connection to prior work—including an additional LPCMCI baseline in experiments, rank-based latent structure discovery methods, and INAR processes—has been expanded.

**Weaknesses:**

1. Novelty of Theorem 4.1 can still be debated. While the authors’ expanded discussion distinguishes their formulation from prior binning-based estimation approaches, the contribution could still benefit from more explicit formal comparison (e.g., showing in what sense their linear representation differs from INAR-based or EM-based formulations beyond the absence of likelihood modeling).

2. Motivation benefits from more discussion. In much of the classical literature on the broader causal discovery problem, the structure within them is not discussed, as the latent confounder are often treated as root nodes affecting the observables. The work is indeed interesting in a theoretical sense, yet, I'd like to question the motivation for latent structure discover: Since these variables are not observed, it is hard to imagine interventions on them, so why is it of interest to practitioners to identify the structure of the latent variables?

3. Assumptions and their implications. The identifiability results depend on assumptions about the Hawkes process and the structure of latent confounder. In which practical scenarios are these assumptions justifiable? On the other hand, in misspecified cases, how poorly is the latent structure discovered? A short illustrative example, even synthetic, where the assumption are valid and one where they fail would greatly benefit the reader.

4. Missing acknowledgement/comparison with recent work on causal discovery in Hawkes processes via compression schemes, e.g., [1,2]

[1] Hlaváčková-Schindler, K., Melnykova, A., & Tubikanec, I. (2024). “Granger causal inference in multivariate Hawkes processes by minimum message length.” JMLR 25(133): 1–26

[2] Jalaldoust, A., Hlaváčková-Schindler, K., & Plant, C. (2022). “Causal Discovery in Hawkes Processes by Minimum Description Length.” AAAI 36(6): 6978–87.

**Questions:**

Please address the weaknesses mentioned above.

---

> ### Author Response · Authors · 2025-11-21
> **Rebuttal by Authors (Part 1/2)**
>
> We thank the reviewer for the insightful comments and valuable feedback. Please see our responses to your concerns point-by-point below.
>
> > **Q1: Novelty of Theorem 4.1 can still be debated. While the authors’ expanded discussion distinguishes their formulation from prior binning-based estimation approaches, the contribution could still benefit from more explicit formal comparison (e.g., showing in what sense their linear representation differs from INAR-based or EM-based formulations beyond the absence of likelihood modeling).**
>
> **A1:** Thank you for your thoughtful comment. In Section 2 of the main paper, we briefly review prior work on Hawkes processes, including both continuous-time and binning-based estimation approaches. **A more extensive discussion is provided in Appendix A due to space limitation of the initial submission.**
>
> **In particular, Appendix A.1 offers a detailed comparison with a representative binning-based method [3], which reconstructs the likelihood of binned event counts by simulating possible within-bin event times via importance sampling and optimizing the objective function.** Such likelihood-based approaches operate directly on aggregated counts and do not provide a structural representation of causal dependencies among the discretized variables themselves.
>
> **In contrast, Theorem 4.1 establishes a direct and explicit structural correspondence between the continuous-time Hawkes process and a linear causal model over the binned variables.** This representation characterizes how past events contribute to future bins through linear operators determined by the excitation kernel, thereby revealing the underlying dependency and causal structure in the discretized domain. This causal–structural interpretation is fundamentally different from prior binning-based estimation methods.
>
> We have further clarified these distinctions in the revised version.
>
> > **Q2: Motivation benefits from more discussion. In much of the classical literature on the broader causal discovery problem, the structure within them is not discussed, as the latent confounder are often treated as root nodes affecting the observables. The work is indeed interesting in a theoretical sense, yet, I'd like to question the motivation for latent structure discover: Since these variables are not observed, it is hard to imagine interventions on them, so why is it of interest to practitioners to identify the structure of the latent variables?**
>
> **A2:** Thank you for your thoughtful comment. Classical causal discovery methods, such as PC [4], typically rely on the causal sufficiency assumption, under which all relevant variables are observed and no latent components influence the system. **In many practical settings, however, this assumption is rarely satisfied, and ignoring latent variables can easily lead to misleading causal conclusions among the observed variables.**
>
> For instance, consider a simple scenario where an latent variable $L_1$ drives three observed variables $O_1$, $O_2$, and $O_3$. **Applying PC [4] directly to the observed data would produce an undirected triangle among $\{O_1, O_2, O_3\}$, despite the fact that none of the observed variables causally influence one another. Intervening on any one of the three would leave the other two unchanged, contradicting the inferred graph.** If the presence of the latent variable is identified, practitioners can correctly interpret such dependency patterns, avoid drawing spurious causal links among the observed subprocesses, and, when possible, consider measuring or approximating the missing component.
>
> **Latent subprocesses also arise naturally in a broad range of real-world applications.** **Examples include unrecorded neuronal populations in neuroscience, unmeasured micro-seismic activity in earthquake modeling, and hidden spreading sources in epidemiology.** In these domains, understanding how latent components interact with observed subprocesses is valuable for revealing hidden mechanisms, guiding the deployment of additional sensors or data collection efforts, and improving downstream tasks such as prediction, anomaly detection, or intervention planning.
>
> We have expanded the motivation in the revised version.

---

> ### Author Response · Authors · 2025-11-21
> **Rebuttal by Authors (Part 2/2)**
>
> > **Q3: Assumptions and their implications. The identifiability results depend on assumptions about the Hawkes process and the structure of latent confounder. In which practical scenarios are these assumptions justifiable? On the other hand, in misspecified cases, how poorly is the latent structure discovered? A short illustrative example, even synthetic, where the assumption are valid and one where they fail would greatly benefit the reader.**
>
> **A3:** Thank you for the thoughtful question. **Our identifiability results rely on (i) the Hawkes specification $\phi_{ij}(s)=a_{ij}w(s)$ with rank faithfulness and (ii) the path-based characterization in Definition 4.4. The first assumption is standard in multivariate Hawkes modeling**, many real-world applications use such kernel shapes with unknown coefficients, and it ensures that second-order statistics encode meaningful structural information.
>
> **Regarding latent confounders, Definition 4.4 is not a modeling assumption but a precise characterization of when a latent subprocess induces rank deficiency in the relevant sub cross-covariance matrices.** These rank conditions describe exactly when a latent subprocess leaves an identifiable signature in the observed data, whether the influence is direct or mediated by intermediate latent subprocesses.
>
> **The iterative discovery algorithm leverages these rank patterns:** each iteration recovers causal relations among currently known subprocesses and detects new latent ones only when their effects are statistically identifiable. **When the path condition is violated (e.g., unequal path lengths or cycles), the corresponding rank deficiency does not appear, and the algorithm naturally terminates without introducing spurious latent subprocesses. In such cases, the non-identifiable higher-level latent structure cannot be recovered, but the identified portion remains meaningful.**
>
> Moreover, we added additional experiments for contrasting scenarios that violate the path condition (see our response to Question 2 of Reviewer hpp2 and the new sensitivity analysis in the appendix Q.3). The results show that the proposed procedure remains practically robust under mild deviations from the symmetric path condition.
>
> > **Q4: Missing acknowledgement/comparison with recent work on causal discovery in Hawkes processes via compression schemes, e.g., [1,2]**
>
> **A4:** Thank you for highlighting these relevant works. The first paper [1] formulates causal discovery in multivariate Hawkes processes as a model-selection task under the minimum description length (MDL) principle, using Monte Carlo methods to approximate the likelihood. The second paper [2] focuses on Hawkes processes with exponential decay kernels and employs a minimum message‐length (MML) criterion to select graphs with the most concise explanation and has a better performance in scenarios with short time horizons.
>
> Both works concentrate on selecting a causal graph by optimizing an information-theoretic criterion (MDL/MML). In contrast, building on the linear autoregressive representation established in Theorem 4.1, we analyze how second-order statistics encode causal dependencies and derive identifiability conditions that allow recovery of latent subprocesses, a problem that MDL/MML formulations do not address. This structural viewpoint further enables a rank-based discovery algorithm, rather than an optimization of a likelihood or message-length criterion.
>
> We have added these two related works into the revised version.
>
> Thank you again for your valuable feedback and hope that all your concerns are properly addressed.
>
>
> **References:**
>
> [1] Jalaldoust, A., Hlaváčková-Schindler, K., & Plant, C. (2022). “Causal Discovery in Hawkes Processes by Minimum Description Length.” AAAI 36(6): 6978–87.
>
> [2] Hlaváčková-Schindler, K., Melnykova, A., & Tubikanec, I. (2024). “Granger causal inference in multivariate Hawkes processes by minimum message length.” JMLR 25(133): 1–26
>
> [3] Leigh Shlomovich, Edward AK Cohen, Niall Adams, and Lekha Patel. Parameter estimation of binned hawkes
> processes. Journal of Computational and Graphical Statistics, 31(4):990–1000, 2022.
>
> [4] Peter Spirtes, Clark Glymour, and Richard Scheines. Causation, prediction, and search. MIT press, 2001.

---

### Official Review · Reviewer_hpp2 · 2025-11-01

**Soundness:** 3
**Presentation:** 2
**Contribution:** 3
**Rating:** 6
**Confidence:** 3

**Summary:**

The paper studies structure learning (causal discovery) for partially observed multivariate Hawkes processes (PO-MHP) and provides the first principled framework that identifies latent sub-processes and recovers causal structure in continuous-time event sequences without prior knowledge.

The authors make a key theoretical contribution (Theorem 4.1) by showing that a continuous-time multivariate Hawkes process can be represented by a discrete-time linear causal model when the event-count data is appropriately binned. They further prove that the low-rank constraints on the cross-covariance matrices induced by the linear representation can be used to (1) detect the presence of latent confounder subprocesses, and (2) identify parent–cause sets and causal edges under explicit path-based conditions (Definitions 4.4, Propositions 4.3/4.5, Theorems 4.7/4.8).

Based on this theoretical foundation, the paper proposes a novel two-phase iterative algorithm where Phase I identifies causal relationships among the currently known (observed and inferred) subprocesses and Phase I discovers new latent confounders via rank tests. The authors also prove that this method guarantees the identifiability of the causal graph. Experiments on both synthetic and real-world datasets show that the proposed method effectively recovers the ground-truth causal graphs, outperforming existing baselines, especially in settings with complex latent structures.

**Strengths:**

1. One of the paper's main strengths is its strong theoretical foundation. Theorem 4.1 is a powerful result, which innovatively establishes a connection between continuous-time Hawkes processes and a discrete-time linear autoregressive representation. Moreover, the *Definition 4.4 + Proposition 4.5 + Theorems 4.7/4.8* that links symmetric path structures to observable rank deficiencies is also original and enables finding latent confounders without prior information of the existence or number of latent subprocesses.

2. The paper addresses a critical challenge, where many previous causal discovery algorithms assume that all relevant variables are observed. This paper instead studies a more realistic and difficult scenario under partial observability, where they propose a novel framework to uncover causal structure with unknown latent subprocesses. It is scientifically important.

3. The proposed two-phase iterative algorithm is a direct and elegant consequence of the theoretical results. The experiments, while concise, are well-designed to validate the paper's core claims. Specifically, the synthetic experiments include multiple graph families, sample sizes, and sensitivity checks.

**Weaknesses:**

1. Strong structural assumptions. Definition 4.4 formalizes the Symmetric Acyclic Path Situation (the observed effects being connected to the latent via paths of equal length and acyclic intermediate latents), which is a somewhat special topology. However, in complex systems intermediate latents can have varying path lengths or additional cross-links, which would break the condition and make that latent unidentifiable by the method.

2. While the paper is theoretically rigorous, it is also extremely dense. The written could be polished with a motivating real world example in Figure 1, then expand the theoretical proof based on this step-by-step real world example. Moreover, some intuitions could be explained before show the theorems and proofs. For example, the transition from 4.2.1 to 4.2.2 could give some intuitions.

3. Only one real world dataset is limited. This small dataset (evaluation on a five-alarm subgraph) can not support the model effectiveness on large and noise real-world systems. Meanwhile, the reported results on Tables 1-4 do not show variance.

**Questions:**

1. How sensitive is the performance of your algorithm to the choice of the discretization interval Δ? Is there a principled way to select an optimal Δ, or is it purely an empirical choice? How does data sparsity affect this choice?

2. Some additional experiments could be added. For example, what if some latent confounders are removed (which violates the condition in Definition 4.4)?

---

> ### Author Response · Authors · 2025-11-21
> **Rebuttal by Authors (Part 1/2)**
>
> We thank the reviewer for the insightful comments and valuable feedback. Please see our responses to your concerns point-by-point below.
>
> > **Q1: How sensitive is the performance of your algorithm to the choice of the discretization interval $\Delta$? Is there a principled way to select an optimal $\Delta$, or is it purely an empirical choice? How does data sparsity affect this choice?**
>
> **A1:** Thank you for your valuable comment. Our method recovers the causal structure of the discretized time-series representation of a multivariate Hawkes process, whose correspondence to the underlying continuous-time process holds when the discretization interval $\Delta$ is chosen below the intrinsic moment-structure scale $\delta$, as stated in Theorem 4.1 and Remark G.7. Within this range, the linear autoregressive model remains valid and does not distort the true parent-child relations.
>
> **To empirically assess sensitivity, we conducted a systematic analysis of the choice of $\Delta$ in Appendix Q.3.** Using an exponential kernel $a_{i,j}e^{(-\beta s)}$ with decay parameter $\beta = 1$, **we observe in Table 1 of Appendix Q.3 that the recovered causal graphs maintain high F1-scores and exhibit stable structure across a range of $\Delta$ values that are small relative to the support of the excitation function.** This indicates that the method is not overly sensitive once $\Delta$ is within the valid operating region.
>
> Regarding how to select $\Delta$ in practice, the true model parameters are typically unknown. As the reviewer notes, data sparsity offers useful guidance: **sparse event sequences correspond to excitation functions with larger effective support, and therefore permit a larger admissible discretization interval.** Combining this intuition with our sensitivity analysis, **a practical strategy is to perform a grid search over candidate $\Delta$ values and choose one from the stable range, where the recovered graphs remain consistent and robust to small perturbations in $\Delta$.**
>
> We have incorporated this practical guidance into the revised paper.
>
> > **Q2: Strong structural assumptions. Definition 4.4 formalizes the Symmetric Acyclic Path Situation (the observed effects being connected to the latent via paths of equal length and acyclic intermediate latents), which is a somewhat special topology. However, in complex systems intermediate latents can have varying path lengths or additional cross-links, which would break the condition and make that latent unidentifiable by the method. Some additional experiments could be added. For example, what if some latent confounders are removed (which violates the condition in Definition 4.4)?**
>
>
> **A2:** Thank you for your thoughtful comment. We would like to clarify that **Definition 4.4 is, strictly speaking, not a modeling assumption but rather a formal characterization of the graphical configurations under which rank deficiency arises in carefully constructed sub–cross-covariance matrices of the observed subprocesses.** These rank conditions, derived under the Hawkes kernel $\phi_{ij}(s) = a_{ij}w(s)$ and rank faithfulness, specify exactly when a latent confounder becomes identifiable from the observable second-order statistics. Rank deficiency occurs not only when a latent subprocess directly affects multiple observed subprocesses (see Eq. (3) and lines 294–316), but also when its influence is mediated through intermediate latents, as illustrated in Figs. 2(c) and 2(d). Definition 4.4 formalizes these path-based configurations.
>
> **Intuitively, the symmetry requirement in Definition 4.4 arises for two related reasons.** First, **if the path from a latent subprocess to an observed subprocess has length $l$, then its contribution to the elements in the relevant row of the coefficient matrix can be written as a product of $l$ integrals of the kernel shape $w(s)$.** Different path lengths generally yield different functional forms, making the corresponding rows linearly independent and thus preventing rank deficiency. Second, **a path of length $l$ induces $l$ leading zeros in the coefficient matrix's row corresponding to the current observed variable.** When different observed subprocesses are connected through paths of different lengths, the resulting rows have different patterns of leading zeros, again destroying the linear dependence needed for rank deficiency. See Eq. (12) in Appendix K. **Symmetric, acyclic paths are therefore exactly the cases in which the latent contributions align sufficiently to be detectable via rank tests.**

---

> ### Author Response · Authors · 2025-11-21
> **Rebuttal by Authors (Part 2/2)**
>
> **Continue on A2:**
>
> **Our discovery procedure is an iterative algorithm** in which each iteration (i) recovers causal relationships among the currently discovered subprocesses (Phase 1) and (ii) detects additional latent subprocesses (Phase 2). The algorithm thus incrementally reconstructs the latent–observed structure. **When latent confounder subprocesses don't have at least two observed effects so as to satisfy Definition 4.4, Phase 2 introduces no new latent subprocesses, and the algorithm halts.** In such misspecified cases, the higher-level latent structure is no longer identifiable, but the recovered part of the graph remains meaningful.
>
> **Following the reviewer’s suggestion, we also conducted experiments that explicitly violate Definition 4.4.** Specifically, starting from Case 2 in Fig. 2(a), we modify the paths between $L_1$ and $O_1$ as follows: **Case A:** we insert one additional intermediate latent subprocess between $L_1$ and $O_1$; **Case B:** we insert two additional intermediate latent subprocesses between $L_1$ and $O_1$. In both cases, the path from $L_1$ to $O_1$ becomes longer than the paths from $L_1$ to the other observed subprocesses, breaking the symmetric path condition in Definition 4.4. The results (mean ± standard deviation over ten runs) are reported below:
>
> Table: Performance under violations of the symmetric path condition in Definition 4.4 (Case 2). Values are mean ± standard deviation.
> | Case   | Precision (std) | Recall (std) | F1-score (std) |
> |--------|------------------|--------------|----------------|
> | Case A | 0.91 (0.14)      | 0.93 (0.11)  | 0.92 (0.13)    |
> | Case B | 0.89 (0.15)      | 0.90 (0.12)  | 0.89 (0.13)    |
>
> **Theoretically, once the path symmetry is broken, $O_1$ should no longer be identifiable as a child of $L_1$ from the rank conditions alone. Interestingly, in our experiments with exponential kernels $a_{i,j}e^{(-\beta s)}$, the performance does not drop dramatically.** This can be partially explained by the fact that (i) products of integrals of the exponential kernel remain proportional to an exponential form, so the functional differences between path lengths are attenuated, and (ii) small differences in the number of leading zeros (e.g., one or two) may be statistically weak in finite samples. As a result, the method still tends to recover $L_1 \rightarrow O_1$ together with the other children of $L_1$ with high probability. **This suggests that, while Definition 4.4 characterizes the ideal identifiability range, the proposed procedure can remain practically robust under mild deviations from the symmetric path condition.** We have added the corresponding results to the revised paper.
>
> > **Q3: While the paper is theoretically rigorous, it is also extremely dense. The written could be polished with a motivating real world example in Figure 1, then expand the theoretical proof based on this step-by-step real world example. Moreover, some intuitions could be explained before show the theorems and proofs. For example, the transition from 4.2.1 to 4.2.2 could give some intuitions.**
>
> **A3:** Thank you for the helpful suggestion. Improving the readability of the paper is important, especially given the technical depth of the results. Figure 1 is intended to offer a conceptual overview of different representations of a Hawkes process and to clarify the role of latent subprocesses. While including a real-world motivating example would further enhance intuition, it is challenging to identify a dataset with known latent structure that simultaneously matches all components required for illustration. **In the revised version, we have strengthened the exposition where possible. In particular, following your suggestion, we expanded the transition between Sections 4.2.1 and 4.2.2 to provide clearer intuition before introducing the formal theorems.**
>
> > **Q4: Only one real world dataset is limited. This small dataset (evaluation on a five-alarm subgraph) can not support the model effectiveness on large and noise real-world systems. Meanwhile, the reported results on Tables 1-4 do not show variance.**
>
> **A4:** Thank you for raising this point. The primary goal of this work is to establish the theoretical identifiability conditions for Hawkes processes with latent subprocesses. The experimental studies, both synthetic and real-world, serve mainly to validate the proposed discovery algorithm in settings where the ground-truth structure is accessible. Extending the evaluation to larger event datasets is an important direction for future research.
>
> Moreover, reporting variance is indeed informative. **For the additional experiments conducted during the rebuttal phase, we now include standard deviation estimates and have integrated these results into the revised version.**
>
> Thank you again for your valuable feedback and hope that all your concerns are properly addressed.

---

### Official Review · Reviewer_ZQpB · 2025-11-06

**Soundness:** 3
**Presentation:** 4
**Contribution:** 3
**Rating:** 6
**Confidence:** 3

**Summary:**

This paper studies the problem of causal discovery in multivariate Hawkes processes (MHPs) with latent confounders. The idea is to represent a MHP with a specific form of excitation functions as a linear autoregressive model over discretised variables. Afterwards, the authors introduce a set of conditions under which the causal structure is identifiable using rank tests on covariance matrices of observed discretised variables.

**Strengths:**

The paper addresses an important and relevant problem, i.e., causal discovery in multivariate Hawkes processes (MHPs) with hidden confounders.
The theoretical contributions provide valuable insights into causal discovery without assuming causal sufficiency in MHPs and represent an important step toward advancing research in this area.

**Weaknesses:**

The main result builds on representing an MHP as a linear autoregressive model through discretization. However, according to Theorem 4.1, this result holds only when the discretization parameter (\Delta) tends to zero. In practice, for small but finite (\Delta), this leads to model mismatch, which can also be observed in the sensitivity analysis with respect to (\Delta) in Table 1.

Moreover, all identifiability results are derived under the assumption that the linear representation holds, i.e., as \Delta -> 0. However, no guidance is provided on how to choose (\Delta) in practice to ensure consistent results.

The identifiability results further rely on an additional assumption that the excitation functions take the form (a_{i,j}w(s)), for example, the exponential decay function a_{i,j}\exp(-\beta s). While this is a common assumption in the MHP literature, it is often extended to cases where the decay rate \beta is also an unknown, node-specific parameter, i.e., (a_{i,j}\exp(-\beta_i s)). Although this may appear to be a minor modification, it is non-trivial to see how the results of this work extend to such more general excitation functions.

The proposed algorithm has exponential complexity, which limits its scalability. As discussed above, its performance is also sensitive to the choice of \Delta. Furthermore, the method relies on rank tests, which typically require large amounts of observational data. This raises a question regarding Figure 4: assuming the experimental setting is favorable to all baseline methods as well as the proposed approach (i.e., with no latent confounders), how would these methods perform with substantially fewer observations, e.g., significantly less than 30,000?

**Questions:**

Please see above comments.

---

> ### Author Response · Authors · 2025-11-21
> **Rebuttal by Authors (Part 1/2)**
>
> We thank the reviewer for the insightful comments and valuable feedback. Please see our responses to your questions point by point below.
>
> > **Q1: The main result builds on representing an MHP as a linear autoregressive model through discretization. However, according to Theorem 4.1, this result holds only when the discretization parameter ($\Delta$) tends to zero. In practice, for small but finite ($\Delta$), this leads to model mismatch, which can also be observed in the sensitivity analysis with respect to ($\Delta$) in Table 1.
> Moreover, all identifiability results are derived under the assumption that the linear representation holds, i.e., as $\Delta$ -> 0. However, no guidance is provided on how to choose ($\Delta$) in practice to ensure consistent results.**
>
>
> **A1:** Thank you for your thoughtful consideration. Our method recovers the causal structure of the discretized time-series representation of a multivariate Hawkes process, whose correspondence to the underlying continuous-time process holds when the discretization width $\Delta$ is chosen below the intrinsic moment-structure scale $\delta$, as stated in Theorem 4.1 and Remark G.7. **Within this range, the linear autoregressive model remains valid and does not distort the underlying parent-child relations.**
>
> In practice, **$\Delta$ only needs to be sufficiently small rather than infinitesimal.** As demonstrated in the sensitivity analysis in Table 1, the recovered causal graph remains high F1-score and exhibit structural stability across a range of $\Delta$ values that are small relative to the support of the excitation function. This indicates that the model mismatch introduced by using a finite $\Delta$ is mild and does not materially affect causal recovery in this relevant region.
>
> To provide practical guidance, **we recommend selecting $\Delta$ via a simple grid search and choosing a value from the stable range, where the recovered structures remain robust to small perturbations in $\Delta$.** We have added this to the revised version and highlighted the corresponding updates accordingly.
>
> > **Q2: The identifiability results further rely on an additional assumption that the excitation functions take the form ($a_{i,j}w(s)$), for example, the exponential decay function $a_{i,j}\exp(-\beta s)$. While this is a common assumption in the MHP literature, it is often extended to cases where the decay rate $\beta$ is also an unknown, node-specific parameter, i.e., ($a_{i,j}\exp(-\beta_i s)$). Although this may appear to be a minor modification, it is non-trivial to see how the results of this work extend to such more general excitation functions.**
>
> **A2:** Thank you for your insightful comment. In our paper, the excitation functions are modeled in the form $a_{i,j}w(s)$. This formulation includes the widely used exponential kernel $a_{i,j} e^{(-\beta s)}$ as a special case, where the decay rate $\beta$ does not need to be known in advance.
>
> Our current analysis requires the kernel shape $w(s)$ to be common across edges so that the discretized autoregressive representation preserves the key relationships needed for identifiability. Extending the framework to more general excitation functions, such as node-specific decay rates of the form $a_{i,j}e^{(-\beta_i s)}$, is an interesting direction for future work, and we have noted this in the revised version of the paper.
>
> > **Q3: The proposed algorithm has exponential complexity, which limits its scalability.**
>
> **A3:**  Thank you for allowing us to clarify this point. Our algorithm is an iterative procedure in which each iteration (i) recovers the causal relationships among the currently discovered subprocesses and (ii) uncovers additional latent subprocesses. Through this progressive refinement, the algorithm incrementally reconstructs the full causal structure. **As detailed in the complexity analysis in Appendix P, the overall computational cost depends not only on the total number of subprocesses (observed and latent) but also on the structural density of the causal graph, which affects the number of statistical tests and iterations required. The exponential complexity mentioned in the paper reflects a worst-case upper bound.** In practical scenarios, causal graphs are typically sparse and dependencies localized, which significantly reduces the number of required iterations and tests.
>
> Moreover, scalability remains a broader challenge shared by many existing causal discovery approaches. Our work represents a first step toward recovering latent structures in Hawkes processes. Developing more scalable algorithms for large-scale data is indeed an important and promising direction for future research. We have pointed it out in the revised version.

---

> ### Author Response · Authors · 2025-11-21
> **Rebuttal by Authors (Part 2/2)**
>
> > **Q4: Furthermore, the method relies on rank tests, which typically require large amounts of observational data. This raises a question regarding Figure 4: assuming the experimental setting is favorable to all baseline methods as well as the proposed approach (i.e., with no latent confounders), how would these methods perform with substantially fewer observations, e.g., significantly less than 30,000?**
>
> **A4:** Thank you for the valuable suggestion. Following your suggestion, we conducted an additional experiment using substantially fewer observations. Specifically, we reduced the number of samples to 5k and evaluated all baseline methods, except Hier.Rank (which assumes no direct edges among observed variables), on Case 1 in Fig. 1(b), where the underlying graph is fully observed and contains no latent confounders. The results (reported below as mean ± standard deviation across ten runs) show that our method remains competitive even with significantly fewer observations.
>
> Table: Performance of all methods on Case 1 (fully observed, no latent confounders) using only 5k samples. Values are mean ± standard deviation.
> | Method             | Precision (std) | Recall (std) | F1-score (std) |
> |--------------------|-----------------|--------------|----------------|
> | Ours (Discrete)    | 0.95 (0.10)     | 0.62 (0.13)  | 0.74 (0.11)    |
> | Ours (Hawkes)      | 0.97 (0.10)     | 0.57 (0.13)  | 0.71 (0.12)    |
> | SHP                | 1.00 (0.00)     | 0.93 (0.13)  | 0.96 (0.08)    |
> | NPHC               | 0.53 (0.27)     | 0.36 (0.10)  | 0.42 (0.09)    |
> | THP                | 0.68 (0.27)     | 0.56 (0.30)  | 0.60 (0.28)    |
> | RLCD               | 0.40 (0.36)     | 0.13 (0.10)  | 0.19 (0.13)    |
> | LPCMCI             | 0.92 (0.13)     | 0.53 (0.10)  | 0.67 (0.10)    |
>
> We have revised the paper to include these results.
>
> We appreciate the reviewer’s valuable feedback and hope that all your concerns are properly addressed.

---

### Meta-Review · Area_Chair_Z6hE · 2026-01-05

**Summary:**

The paper addresses the problem of causal structure discovery in multivariate Hawkes processes under partial observability and latent confounding. There is a broad agreement that the problem is important, and the core theory is sound. The two phase iterative algorithm is also noted to be positive. The general feedback on writing is also positive.

There were some concerns w.r.t to the delta parameter that the authors have addressed in their rebuttal. There were some further concerns about the modeling assumptions and from the feedback, it appears that the authors have addressed this as well. There is clearly some scope for better validation but the reviewers have weighed that aspect a bit lower.

**Reviewer Concerns:**

The authors have added an additional experiment. Most of teh other concerns have been adequately addressed by the authors.

**Reviewer Scores:**

I feel like the reviewers would either hold their scores or increase them marginally after the responses.

---

### Decision · Program_Chairs · 2026-01-26

Accept (Oral)